# Identification of Paired-related Homeobox Protein 1 as a key mesenchymal transcription factor in pulmonary fibrosis

Emmeline Marchal-Duval[1†], Méline Homps-Legrand[1†], Antoine Froidure[1,2], Madeleine Jaillet[1], Mada Ghanem[1,3], Deneuville Lou[1,3], Aurélien Justet[1,3], Arnaud Maurac[1], Aurelie Vadel[1], Emilie Fortas[1], Aurelie Cazes[1,4], Audrey Joannes[1,5], Laura Giersh[1], Herve Mal[6], Pierre Mordant[1,7], Tristan Piolot[8], Marin Truchin[9], Carine M Mounier[9,10], Ksenija Schirduan[11], Martina Korfei[12], Andreas Gunther[12], Bernard Mari[9], Frank Jaschinski[11], Bruno Crestani[1,3], Arnaud A Mailleux[1]*

[1]Université Paris Cité, Inserm, Physiopathologie et épidémiologie des maladies respiratoires, Paris, France; [2]Université Paris Cité, Inserm, Physiopathologie et épidémiologie des maladies respiratoires, Brussels, Belgium; [3]Assistance Publique des Hôpitaux de Paris, Hôpital Bichat, Service de Pneumologie A, FHU APOLLO, Paris, France; [4]Assistance Publique des Hôpitaux de Paris, Hôpital Bichat, Département d'Anatomopathologie, FHU APOLLO, Paris, France; [5]Univ Rennes, Inserm, EHESP, Irset (Institut de recherche en santé, environnement et travail) - UMR_S 1085, Rennes, France; [6]Assistance Publique des Hôpitaux de Paris, Hôpital Bichat, Service de Pneumologie et Transplantation, FHU APOLLO, Paris, France; [7]Assistance Publique des Hôpitaux de Paris, Hôpital Bichat, Service de Chirurgie Thoracique et Vasculaire, FHU APOLLO, Paris, France; [8]Plate-forme Orion, CIRB, Collège de France/CNRS-UMR7241/INSERM-U1050, PSL Research University, Paris, France; [9]Université Côte d'Azur, CNRS, IPMC, FHU-OncoAge, Valbonne, France; [10]CYU Université, ERRMECe(EA1391), Neuville sur Oise, France; [11]Secarna Pharmaceuticals GmbH & Co. KG, Planegg, Germany; [12]Department of Internal Medicine II, University of Giessen-Marburg Lung Center, Justus-Liebig University Giessen, Giessen, Germany

*For correspondence: arnaud.mailleux@inserm.fr

[†]These authors contributed equally to this work

**Abstract** Matrix remodeling is a salient feature of idiopathic pulmonary fibrosis (IPF). Targeting cells driving matrix remodeling could be a promising avenue for IPF treatment. Analysis of transcriptomic database identified the mesenchymal transcription factor PRRX1 as upregulated in IPF. PRRX1, strongly expressed by lung fibroblasts, was regulated by a TGF-β/PGE2 balance in vitro in control and IPF human lung fibroblasts, while IPF fibroblast-derived matrix increased *PRRX1* expression in a PDGFR-dependent manner in control ones. PRRX1 inhibition decreased human lung fibroblast proliferation by downregulating the expression of S phase cyclins. PRRX1 inhibition also impacted TGF-β driven myofibroblastic differentiation by inhibiting SMAD2/3 phosphorylation through phosphatase PPM1A upregulation and TGFBR2 downregulation, leading to TGF-β response global decrease. Finally, targeted inhibition of *Prrx1* attenuated fibrotic remodeling in vivo with intra-tracheal antisense oligonucleotides in bleomycin mouse model of lung fibrosis and ex vivo using human and mouse precision-cut lung slices. Our results identified PRRX1 as a key mesenchymal transcription factor during lung fibrogenesis.

## Editor's evaluation

This manuscript is of interest to scientists in the field of tissue injury and repair. It provides novel molecular mechanisms of a transcription factor, Prrx1, in fibroblast activation following lung injury. Overall, the work suggests that PRRX1 plays a functional role downstream of TGFb1 to elicit some aspects of the fibrotic response and that PRRX1 could represent an important therapeutic target to treat fibrosis.

## Introduction

Chronic remodeling is a key feature of many Human diseases associated with aging. In particular, chronic respiratory diseases, including lung fibrosis, are a major and increasing burden in terms of morbidity and mortality (*Martinez et al., 2017*). Idiopathic pulmonary fibrosis (IPF) is the most common form of pulmonary fibrosis. IPF is defined as a specific form of chronic, progressive fibrosing interstitial pneumonia of unknown cause. IPF patients have an overall median survival of 3–5 years, although current antifibrotics (pirfenidone and nintedanib) slow lung function decline and may improve survival (*Froidure et al., 2020*; *Martinez et al., 2017*).

According to the current paradigm, IPF results from progressive alterations of alveolar epithelial cells leading to the recruitment of mesenchymal cells to the alveolar regions of the lung with secondary deposition of extracellular matrix, and destruction of the normal lung structure and physiology. IPF develops in a susceptible, aging individual and is promoted by interaction with environmental agents such as inhaled particles, tobacco smoke, inhaled pollutants, viral and bacterial agents (*Froidure et al., 2020*; *Martinez et al., 2017*).

In any given cell, a set of transcription factors is expressed and works in concert to govern cellular homeostasis and function. Dysregulation of transcriptional networks may therefore account to aberrant phenotypic changes observed in lung fibrosis, as reviewed in *Froidure et al., 2020*. Among the multifaceted tissue cellular 'ecosystem', cells of mesenchymal origin such as fibroblasts are the main cellular components responsible for tissue remodeling during normal and pathological lung tissue repair (*Fernandez and Eickelberg, 2012*; *Martinez et al., 2017*). Thus, targeting master transcription factors preferentially expressed in fibroblast could be a promising avenue for IPF treatment.

Using an in silico approach, we identified the 'Paired Related Homeobox Protein-1' (*PRRX1*) gene as a potential candidate for transcriptional regulation differently modulated in IPF compared to control lungs. The *PRRX1* mRNA generates by alternative splicing two proteins, *PRRX1a* (216 aa) and *PRRX1b* (245 aa) that differ at their C-terminal parts. Functional in vitro studies suggested that PRRX1a promoted transcriptional activation whereas PRRX1b may act rather as a transcriptional repressor in mouse cells (*Norris and Kern, 2001*).

*Prrx1* is implicated in the regulation of mesenchymal cell fate during embryonic development. PRRX1 is essential for fetal development as *Prrx1*[-/-] mice present severe malformation of craniofacial, limb, and vertebral skeletal structures (*Martin et al., 1995*). *Prrx1*[-/-] mice also display hypoplastic lungs with severe vascularization defects and die soon after birth (*Ihida-Stansbury et al., 2004*). PRRX1 function is not restricted to embryogenesis. It has been also shown that PRRX1 was a stemness regulator (*Shimozaki et al., 2013*), involved in adipocyte differentiation (*Du et al., 2013*), epithelial tumor metastasis and pancreatic regeneration (*Fazilaty et al., 2019*; *Ocaña et al., 2012*; *Reichert et al., 2013*) as well as liver fibrosis (*Gong et al., 2017*). PRRX1 transcription factors are also at the center of the network coordinating dermal fibroblast differentiation (*Tomaru et al., 2014*). A *Prrx1*-positive fibroblast subpopulation was also characterized as pro-fibrotic in the mouse ventral dermis (*Currie et al., 2019*; *Leavitt et al., 2020*) as well as modulating mouse skin inflammation during atopic dermatitis (*Ko et al., 2022*). This *Prrx1* mesenchymal subpopulation was also recently involved in scarless repair within the oral mucosa (*Ko et al., 2023*) in mice. In the adult mouse lung, *Prrx1* mRNA expression was associated with a sub-population of matrix fibroblast using single-cell RNA sequencing (*Xie et al., 2018*). However, whether and how PRRX1 plays a functional role in lung fibrogenesis still remains elusive. Given its central position in fibroblast transcriptional network, we hypothesized that PRRX1 transcription factors are important drivers of the fibroblast phenotype in IPF, promoting the development/progression of fibrosis.

## Results

### Identification of *PRRX1* isoforms as mesenchymal transcription factors associated with IPF

Since mesenchymal cells are thought to be one of the major effector cells during fibrosis (*Fernandez and Eickelberg, 2012*; *Martinez et al., 2017*), we sought to identify mesenchymal transcription factors associated with IPF in patients. We screened three curated publicly available transcript microarray databases from NCBI GEO (GDS1252, GDS4279, GDS3951) for transcription factor expression in IPF and control whole lung samples. Among the 210 common genes upregulated at the mRNA level in all three IPF lung datasets compared to their respective control ones (*Figure 1A* and *Supplementary file 1*), 12 genes were annotated as transcription factors (*Figure 1A*) after gene ontology analysis. One of these transcription factors, *PRRX1* appeared as an appealing candidate since this gene was previously associated with mesenchymal cell fate during embryogenesis (*Martin et al., 1995*) and is required for proper lung development (*Ihida-Stansbury et al., 2004*). In addition, *PRRX1* mRNA was upregulated in a fourth transcriptome dataset comparing 'rapid' and 'slow' progressor subgroups of IPF patients (*Selman et al., 2007*). None of those transcriptome datasets discriminated *PRRX1* isoforms, namely *PRRX1a* and *PRRX1b*.

First, we confirmed that both *PRRX1* isoforms were upregulated in Human IPF lungs at the mRNA and protein levels (*Figure 1A–C*). Immunoblot revealed that PRRX1a protein (210 aa) was the main PRRX1 isoform expressed in control and IPF lungs. We also investigated PRRX1 expression pattern by immunohistochemistry in control and IPF Human lung tissue sections (the antibody recognized both PRRX1 isoforms, see *Figure 1D*). PRRX1 staining revealed an accumulation of PRRX1-positive cells in IPF compared to control lungs (*Figure 1—figure supplement 1*). Additional lineage markers were also investigated such as Vimentin (mesenchyme marker), ACTA2 (myofibroblast / smooth muscle marker) and CD45 (hematopoietic lineage) as shown in *Figure 1—figure supplement 1*. PRRX1-positive cells were not detected in the distal alveolar space and in the bronchiolar epithelium of control lung (*Figure 1D*). Nevertheless, PRRX1 nuclear staining was observed in mesenchymal cells (Vimentin-positive but ACTA2 and CD45-negative cells) within peri-vascular and peri-bronchiolar spaces (*Figure 1D* and *Figure 1—figure supplement 1*). In IPF patients, nuclear PRRX1-positive cells were mainly detected in the fibroblast foci (also observed by others *Yeo et al., 2018*) which are the active sites of fibrogenesis (*Fernandez and Eickelberg, 2012*; *Martinez et al., 2017*) and in scattered mesenchymal cells within the remodeled / fibrotic lung areas (*Figure 1D*). Those PRRX1-positive cells were all Vimentine positive and CD45 negative but only some were expressing ACTA2 (*Figure 1—figure supplement 1*). To confirm the identity of PRRX1 expressing cells in the lung as fibroblasts, we took advantage of recently published single-cell transcriptomic analysis (scRNAseq) performed using lung samples (*Habermann et al., 2020*). *PRRX1* mRNA expression was restricted to the fibroblast / mesenchymal lung cell lineages in either lung transplant donors or recipients with pulmonary fibrosis (*Figure 2A–B* and *Figure 2—figure supplement 1*). In control Human lung, *PRRX1* expression was mainly associated with adventitial fibroblasts. In IPF, *PRRX1* was also expressed in fibrosis-associated fibroblast cell populations (*HAS1* high and *PLIN2* +fibroblasts as well as *ACTA2* +myofibroblasts as defined by *Habermann et al., 2020*). In vitro, both isoforms, *PRRX1a* and *PRRX1b* mRNA were also found to be strongly expressed by primary control lung fibroblasts compared to primary alveolar epithelial type 2 cells (AECII) and alveolar macrophages (*Figure 2C*) by quantitative PCR (qPCR). In addition, *PRRX1a* and *–1b* levels were increased in IPF primary lung fibroblasts compared to control ones only at the mRNA level as assayed by qPCR and western blot (*Figure 2D–E*). By immunofluorescence, PRRX1 was detected at the protein level in the nuclei of both control and IPF fibroblasts cultured in vitro (*Figure 2F*).

### *PRRX1* isoforms expression in primary lung fibroblasts is tightly regulated by growth factors and extracellular matrix stiffness in vitro

To better understand the regulation of PRRX1 isoforms in lung fibroblasts, we first assayed the effects of factors known to regulate lung fibroblast to myofibroblasts differentiation on the expression of both PRRX1 isoforms in control and IPF primary lung fibroblasts. 'Transforming growth factor beta 1' (TGF-β1, 1 ng/ml) treatment which triggers myofibroblastic differentiation (*Fernandez and Eickelberg, 2012*) was associated with a decrease in the expression level of both *PRRX1* isoforms at the mRNA

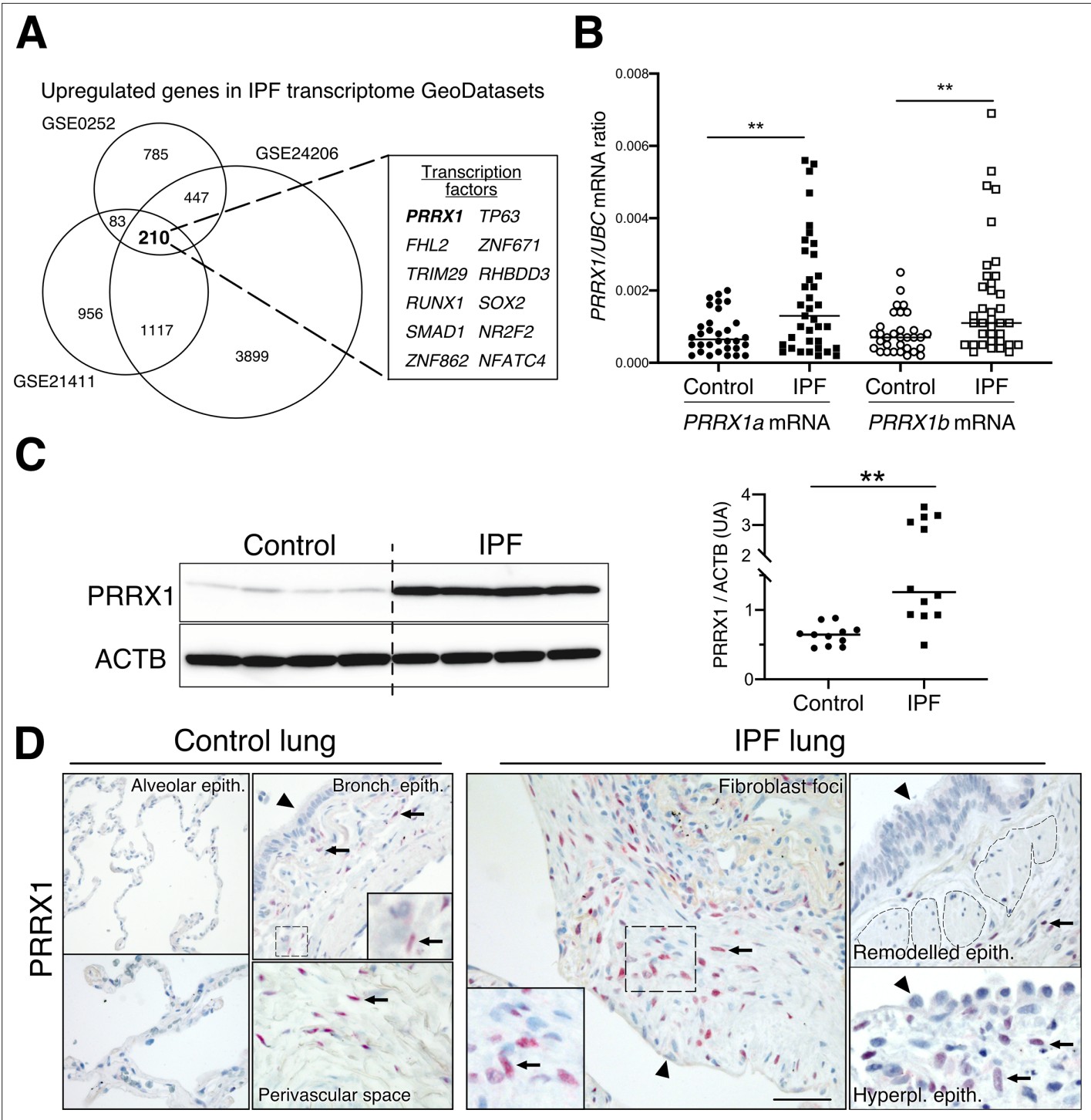

**Figure 1.** Identification of PRRX1 as a transcription factor reactivated in IPF lung. (**A**) Venn diagram showing the number of genes up-regulated in three IPF lung Transcript microarray databases compared to controls (NCBI GEO GDS1252, GDS4279, GDS3951). Among the 210 common upregulated genes in all three datasets, 12 genes were annotated as transcription factors (see embedded table in the figure, PRRX1 is in bold). (**B**) Dot plots with median showing the mRNA expression of *PRRX1a* and *PRRX1b* isoforms in control (circle, n=32) and IPF (square, n=37) whole lung homogenates. (**C**) Immunoblot showing PRRX1 expression in control and IPF whole lung homogenates. ACTB was used as loading control. The quantification of PRRX1 relative expression to ACTB in control (circle, n=11) and IPF (square, n=12) is displayed as dot plot with median on the right. (**D**) Representative immunohistochemistry images (n=5 per group) showing PRRX1 staining (red) in control (left panels) and IPF (right panels). Nuclei were counterstained with hematoxylin. Note the absence of PRRX1 staining in the alveolar and bronchiolar epithelium (arrow head). PRRX1-positive cells were only detected in the peri-bronchiolar and peri-vascular spaces (arrows) in control lungs (left panels). In IPF, PRRX1-positive cells (arrow) were detected in the

*Figure 1 continued on next page*

*Figure 1 continued*

remodeled/fibrotic area (right panels). Note that epithelial cells (arrow head) and bronchiolar smooth muscle cells (dashed areas) are PRRX1 negative. The high-magnification pictures match the dashed boxes displayed in the main panels. Scale bar: 80 μm in low magnification images and 25 μm in high-magnification ones. *Abbreviations: epithelium. (epith); bronchiolar. (bronch); Hyperpl. (hyperplastic).* Mann Whitney U test, *p≤0.05, **p≤0.01.

The online version of this article includes the following source data and figure supplement(s) for figure 1:

**Source data 1.** The zip folder contains the data presented in panels B-C (in an Excel document).

**Figure supplement 1.** Co-expression of PRRX1 with Vimentin and ACTA2 in control and IPF lungs by immunochemistry.

**Figure supplement 1—source data 1.** The zip folder contains the data presented in the panel A (in an Excel document).

level (*Figure 3A*) at 48 hr. A partial inhibitory effect was confirmed at the protein level by western blot and immunofluorescence at 48 hr (*Figure 3—figure supplement 1*). Time course analysis of PRRX1 isoform expression levels (at 4, 8, 16, 24, and 48 h) showed that PRRX1 downregulation was a late event after TGF-β1 stimulation in lung fibroblasts. After TGF-β1 treatment, *PRRX1a* and *PRRX1b* levels were decreased at the mRNA level only in control lung fibroblasts at 24 hr but at both mRNA

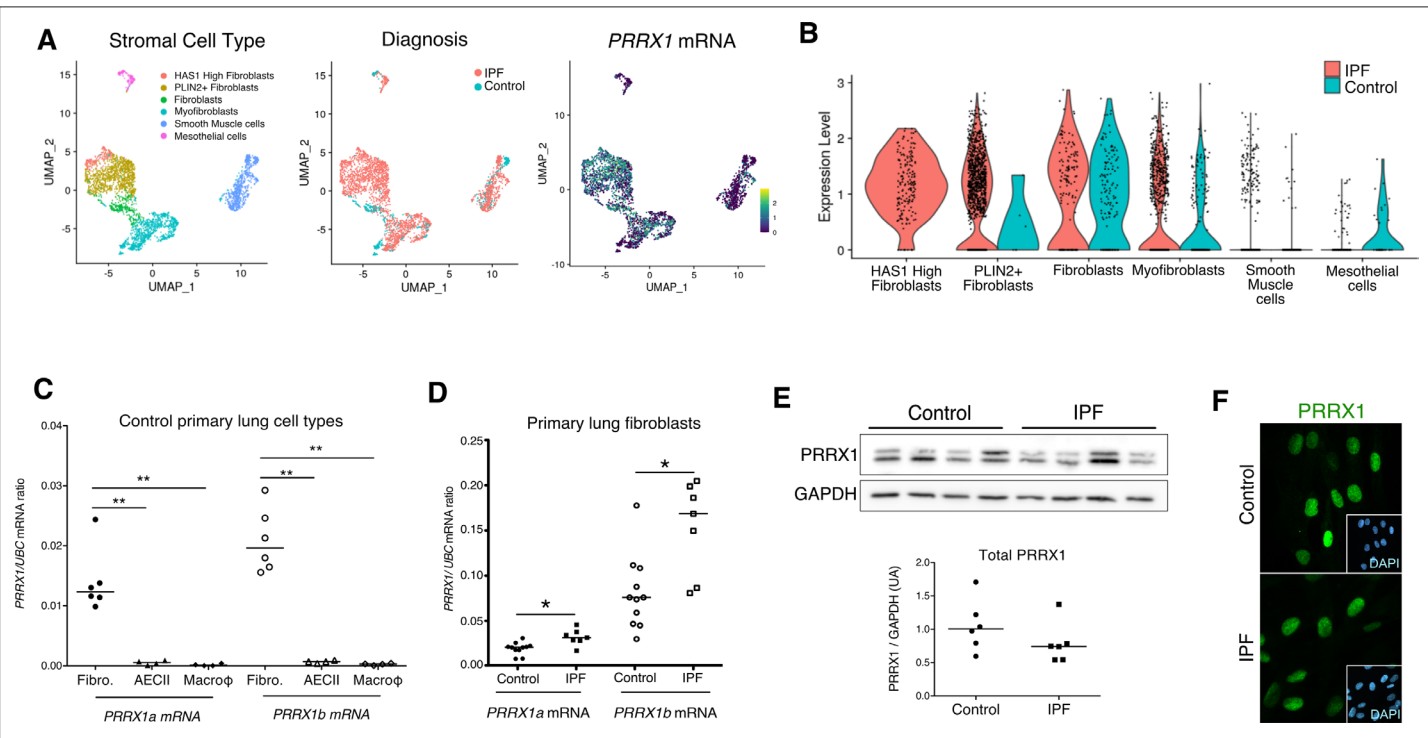

**Figure 2.** PRRX1 is a mesenchymal transcription factor upregulated in primary Human lung IPF fibroblasts. (**A**) UMAP plots describing the distribution of *PRRX1* expressing cells in stromal subpopulations using Banovich / Kropski's dataset (*Adams et al., 2020*; *Habermann et al., 2020*). The labeling of each cell cluster is showed on the left, the diagnosis in the middle and *PRRX1* relative expression on the right. (**B**) Violin plots visualizing *PRRX1* mRNA expression from (**A**) in each cell type stratified by disease states (control lung cell types in blue and IPF ones in red). Note that *PRRX1* mRNA expression was associated with stromal clusters in both conditions and expressed in IPF associated subpopulation (*HAS1* high and *PLIN2+* Fibroblasts). (**C**) Dot plots with median showing the mRNA expression of *PRRX1a* (black) and *PRRX1b* (white) isoforms in primary Human lung fibroblasts (circle, n=6), alveolar epithelial cells (triangle, n=4) and alveolar macrophages (diamond, n=4). (**D**) Dot plots with median showing the mRNA expression of *PRRX1a* (black) and *PRRX1b* (white) isoforms in control (circle, n=11) and IPF (square, n=7) primary Human lung fibroblasts. (**E**) Immunoblot showing PRRX1 expression in control and IPF primary Human lung fibroblasts. GAPDH was used as loading control. The quantification of PRRX1 relative expression to GAPDH in control (circle, n=6) and IPF (square, n=6) lung fibroblasts is displayed as dot plot with median below. (**F**) Representative Immunofluorescence images (n=8 per group) showing PRRX1 staining (green) in control (top panel) and IPF (bottom panel) fibroblasts. Nuclei were counterstained with DAPI (inserts in main panels). Scale bar 20 μm in main panels and 40 μm in inserts. *Abbreviations: fibroblasts (fibro); alveolar epithelial cells (AECII); alveolar macrophages (macroφ).* Mann Whitney U test, *p≤0.05.

The online version of this article includes the following source data and figure supplement(s) for figure 2:

**Source data 1.** The zip folder contains the data presented in panels C-E (in an Excel document).

**Figure supplement 1.** *PRRX1* expression profiles at single-cell resolution using the 'IPF Cell Atlas' web database (http://ipfcellatlas.com/).

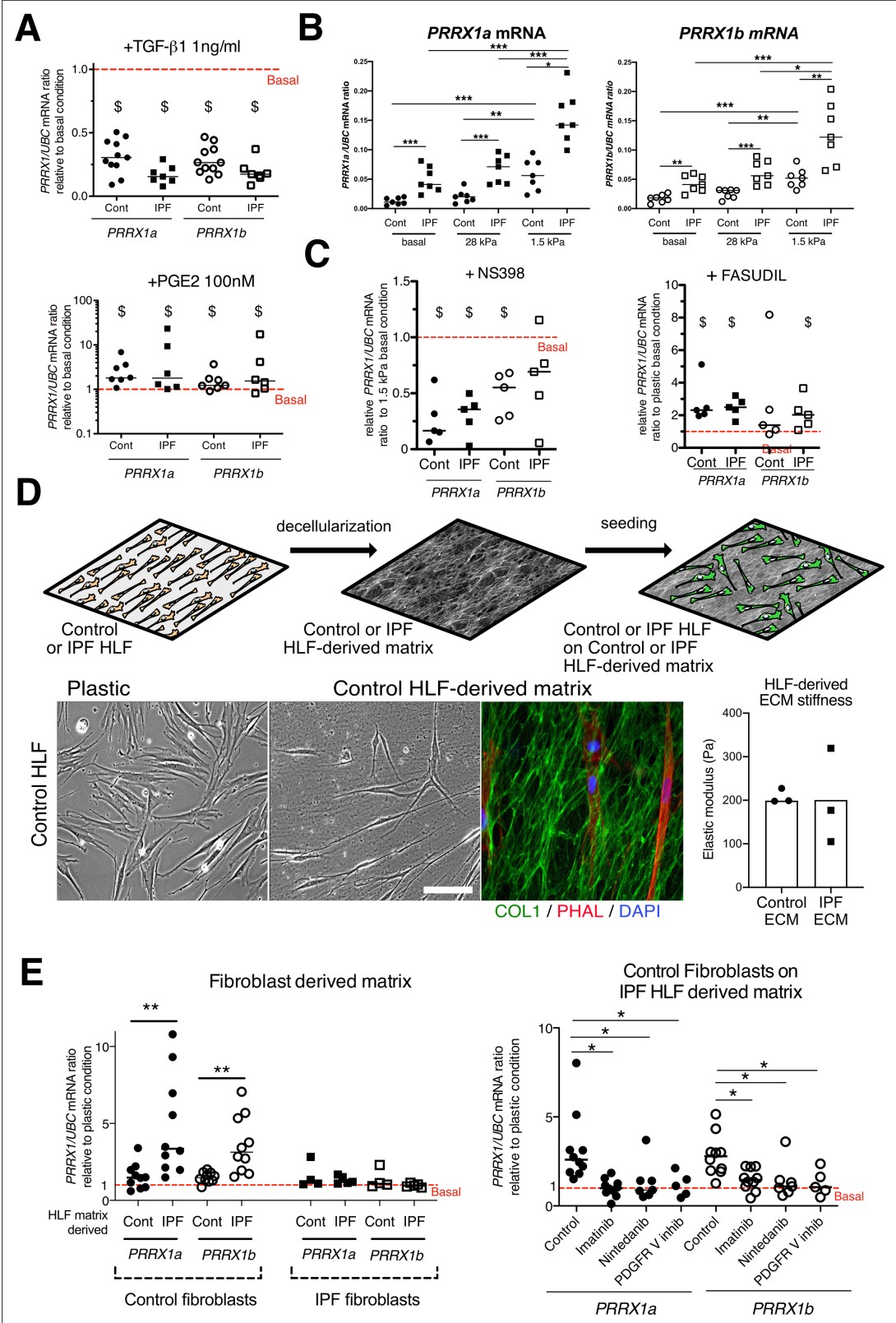

**Figure 3.** PRRX1 is modulated by growth factors and matrix environment. (**A**) Dot plots with median showing the mRNA expression of *PRRX1a* (black) and *PRRX1b* (white) isoforms in control (circle) and IPF (square) primary Human lung fibroblasts stimulated for 48 hr with TGF-β1 (left; n=11 control HLF, n=7 IPF HLF), PGE2 (middle; n=7 control HLF, n=6 IPF HLF) compared to basal condition (red dashed line). (**B**) Dot plots with median showing the mRNA expression of *PRRX1a* (left panel) and *PRRX1b* (right panel) isoforms in control (circle) and IPF (square) lung fibroblasts (n=7 per group) cultured on

*Figure 3 continued on next page*

*Figure 3 continued*

plastic (basal) or on stiff (28 kPa) and soft (1.5 kPa) substrate. (**C**) Dot plots with median showing the mRNA expression of *PRRX1a* and *PRRX1b* isoforms in control and IPF lung fibroblasts (n=5) stimulated 48 hr with NS398 (left), or Fasudil (right) compared to basal condition. (**D**) Summary sketch of Fibroblast-derived matrix experiments (upper part). Lower part: representative phase contrast pictures of control primary lung fibroblasts on plastic (left) or seeded in a control fibroblast-derived matrix (middle) and immunofluorescence pictures (right) of Collagen 1 (green) revealing the HLF-derived matrix and actin fibers stained with Phalloidin (red). Nuclei were counterstained with DAPI (blue). Right panel: column with mean and individual values showing the mean stiffness (elastic modulus) of control (circle) or IPF (square) fibroblast-derived matrix (n=3 per group). (**E**) Left panel: dot plots with median showing the mRNA expression of *PRRX1a* and *PRRX1b* isoforms in control or IPF fibroblasts seeded on control (n=10 per control) or IPF (n=4 control HLF and n=5 IPF HLF) derived matrix compared to basal condition. Right panel: dot plots with median showing the mRNA expression of *PRRX1a* and *PRRX1b* isoforms in control fibroblasts cultured on IPF derived matrix and stimulated with Imatinib (n=10), Nintedanib (n=7) or PDGFR V inhibitor (n=5) compared to basal condition. (Scale bar: 30 µm in phase contrast pictures and 15 µm in the immunofluorescence ones) *Abbreviations: Control (Cont), Human lung fibroblast (HLF), Elastic/Young modulus (EM), PHAL (Phalloidin), COL1 (Collagen 1).* Mann Whitney U test, *p≤0.05 **p≤0.01; Wilcoxon signed-rank test $ p≤0.05.

The online version of this article includes the following source data and figure supplement(s) for figure 3:

**Source data 1.** The zip folder contains the data presented in panels A-C and E (in an Excel document).

**Figure supplement 1.** Regulation of PRRX1 protein expression by growth factors as assayed by qPCR, immunoblots and immunofluorescence.

**Figure supplement 1—source data 1.** The zip folder contains the data presented in panels A-F (in an Excel document).

and protein levels in control and IPF lung fibroblast only at 48 hr (see *Figure 3A* and *Figure 3—figure supplement 1*). Conversely, prostaglandin E₂ (PGE₂, 100 nM) treatment which decreases myofibroblastic differentiation (*Fernandez and Eickelberg, 2012*) was associated with an increase of *PRRX1* isoforms mRNA and protein in both control and IPF fibroblasts (*Figure 3A* and *Figure 3—figure supplement 1*) at 48 hr. These data indicate that PRRX1 isoform expression is controlled by a TGF-β/PGE2 balance in lung fibroblasts.

Concomitantly with an aberrant growth factors/chemokine secretory profile, lung fibrosis is also characterized by local matrix stiffening, which plays a key role in IPF physiopathology (*Booth et al., 2012*). Previous studies showed that increasing matrix stiffness strongly suppressed fibroblast expression of *PTGS2* (*Liu et al., 2010*), a key enzyme in PGE₂ synthesis, and increased Rho kinase (ROCK) activity (*Zhou et al., 2013*), contributing to myofibroblastic differentiation. Control and IPF primary lung fibroblasts were cultured on fibronectin-coated glass (elastic/Young's modulo in the GPa range) or hydrogel substrates of discrete stiffness, spanning the range of normal (1.5 kPa) and fibrotic (28 kPa) lung tissue (*Booth et al., 2012*). We confirmed that soft (1.5 kPa) substrate culture condition did increase *PTGS2* mRNA level compared to stiff/glass control condition in both control and IPF fibroblasts (data not shown). The expression levels of both *PRRX1* TFs isoforms mRNA were also increased on soft/normal 1.5 kPa stiffness substrate (*Figure 3B*) compared to stiff substrates (Glass and 28 kPa culture conditions) in control and IPF fibroblasts. In addition, the increase of both *PRRX1* isoform mRNA levels was still observed in IPF fibroblasts compared to control ones when cultured on 1.5 kPa or 28 kPa hydrogels (*Figure 3B*). Treatment with NS398 (10 µg/ml), a specific PTGS2 inhibitor abrogated the *PRRX1* TFs increase on soft substrate (*Figure 3C*). Conversely, inhibition of mechanosensitive signalling with Fasudil (35 µM), an inhibitor of ROCK1 and ROCK2 (*Zhou et al., 2013*), induced *PRRX1* TFs mRNA expression in both control and IPF fibroblasts grown on glass/stiff substrate (*Figure 3C*). Collectively, these data indicate that PRRX1 expression is tightly controlled by extra-cellular matrix stiffness through a PTGS2/ROCK activity balance.

## IPF fibroblast-derived matrix increased PRRX1 expression in control fibroblasts in a PDGFR dependent manner

To better appreciate the regulation of PRRX1 expression in a complex environment, we cultured lung fibroblasts in a fibroblast-derived 3D ECM (*Castelló-Cros and Cukierman, 2009*). This 3D in vitro model will enable the study of how the interactions between lung fibroblasts and control or IPF stroma may affect PRRX1 expression levels. In this model, control and IPF fibroblasts were maintained in high-density culture to generate thick matrices that were extracted with detergent at alkaline pH to remove cellular contents (*Figure 3D*). After treatment, a 3D ECM is left behind that is both intact and cell-free (*Castelló-Cros and Cukierman, 2009*). The elastic modulus of this ECM, as measured by atomic force microscopy, is approximately 200 Pa (*Figure 3D*). Thus, the stiffness of the ECM produced by both control and IPF fibroblasts falls within a similar range. When

seeded onto 3D ECM derived from IPF fibroblasts, we observed an upregulation of *PRRX1a* and *–1b* TF mRNA expression in control fibroblasts compared to either plastic culture or control fibroblast derived 3D ECM. Meanwhile, *PRRX1a* and *–1b* mRNA expression levels were stable in IPF fibroblasts seeded either on control or IPF fibroblast derived 3D ECM compared to plastic culture (*Figure 3E*). These results suggested that the ECM derived from IPF fibroblasts (namely its origin) regulated the expression of *PRRX1* transcription factors in control lung fibroblast independently of potential differences in stiffness in this assay. To gain a deeper understanding of the cellular processes and signaling pathways involved, we treated control fibroblasts seeded on IPF fibroblast-derived matrix with two tyrosine kinase protein inhibitors known for their anti-fibrotic properties (*Grimminger et al., 2015*): Imatinib (10 µg/ml) and Nintedanib (10 nM). Notably, Nintedanib is one of two drugs currently approved for IPF treatment (*Martinez et al., 2017*). Both inhibitors reverted the effect of IPF fibroblast-derived matrix upon *PRRX1a* and *PRRX1b* mRNA levels (*Figure 3E*). Interestingly, amongst their multiple targets (*Fernandez and Eickelberg, 2012*; *Martinez et al., 2017*), Imatinib and Nintedanib are known to both inhibit PDGFR. Treatment with a specific PDGFR inhibitor (PDGFR V *Furuta et al., 2006*) at the nanomolar range (10 nM) did confirm the PDGFR-dependency of this effect of IPF fibroblast-derived ECM on *PRRX1a* and *–1b* mRNA levels in control fibroblasts (*Figure 3E*). Our findings indicate that constitutive components of the microenvironment may play a role in controlling PRRX1 expression in control fibroblasts (through PDGFR activation), regardless of local stiffness.

## PRRX1 TF isoforms promote fibroblast proliferation

Since PRRX1 expression is strongly associated with fibroblasts in IPF, we next investigated whether PRRX1 TFs may drive the phenotype of primary lung fibroblasts. The involvement of PRRX1 was studied in vitro by using siRNA targeting both *PRRX1a* and *PRRX1b* isoforms (loss of function).

First, we investigated the effects of *PRRX1* TFs knock down using two different siRNA sequences (see *Figure 4A–B*). Knockdown of *PRRX1* TFs significantly decreased primary lung fibroblast proliferation in complete growth medium after 72 hr (*Figure 4C*) as compared to the control siRNA. Cell cycle analysis revealed a significant decrease in S phase concomitantly with an increase in G1 phase, suggestive of a G1/S arrest in control and IPF lung fibroblasts treated with *PRRX1* siRNA (*Figure 4D* and *Figure 4—figure supplement 1*). A decreased BrdU incorporation in *PRRX1* siRNA-treated control and IPF lung fibroblasts also supported a potential G1/S arrest in these cells compared to control siRNA-treated lung fibroblasts at 72 hr (*Figure 4—figure supplement 1*). This potential G1/S arrest was also associated with a strong decrease in *CCNA2* and *CCNE2* mRNA expression after 72 hr (*Figure 4E*). These two cyclins play a key role in the replicative S phase during the cell cycle (*Lim and Kaldis, 2013*). To further characterize the impact of *PRRX1* TF inhibition on cell cycle progression, we performed a FACS analysis of KI67 expression in primary control and IPF lung fibroblasts transfected with *PRRX1* siRNA sequences for 72 hr compared to control siRNA. KI67 protein is usually present during all active phases of the cell cycle, but is absent from resting cells in G0 (*Sobecki et al., 2017*). *PRRX1* inhibition strongly decreased the number of KI67 (MKI67; official name) positive cells in control and IPF lung fibroblasts (*Figure 4F* and *Figure 4—figure supplement 1*). Of note, *MKI67* expression was also decreased at the mRNA level in control and IPF lung fibroblasts treated with *PRRX1* siRNA (*Figure 4—figure supplement 1*). Next, we used a chromatin immunoprecipitation approach (ChIP) to assay a possible direct regulatory effect of PRRX1 upon *CCNA2, CCNE2* and *MKI67* gene loci in primary normal Human lung fibroblasts (NHLF). We observed an enrichment of PRRX1 binding at the vicinity of PRRX1 response element [2](*PRE*) identified in the *CCNA2, CCNE2* and *MKI67* promoter regions, suggesting that those genes could be direct PRRX1 TFs target genes. Meanwhile, no PRRX1 binding was detected at the *GAPDH* transcription starting site (TSS); devoid of PRE (*Figure 4G*).

We also assayed the effect of *PRRX1* knock down on the mRNA expression of *CDKN2A* (p16), *CDKN1A* (p21), and *TP53*, major negative regulators of cell cycle also associated with cellular senescence (*Lim and Kaldis, 2013*). The expression of all three cell cycle inhibitors was increased only at the mRNA level in both control and IPF lung fibroblasts treated with *PRRX1* siRNA compared to control siRNA as assayed by qPCR and western blot (see *Figure 4—figure supplement 1* and data not shown). This cell cycle arrest in control and IPF lung fibroblasts treated with *PRRX1* siRNA was not associated with an increase in β-Galactosidase activity, a senescence marker, compared to cells transfected with control siRNA (data not shown).

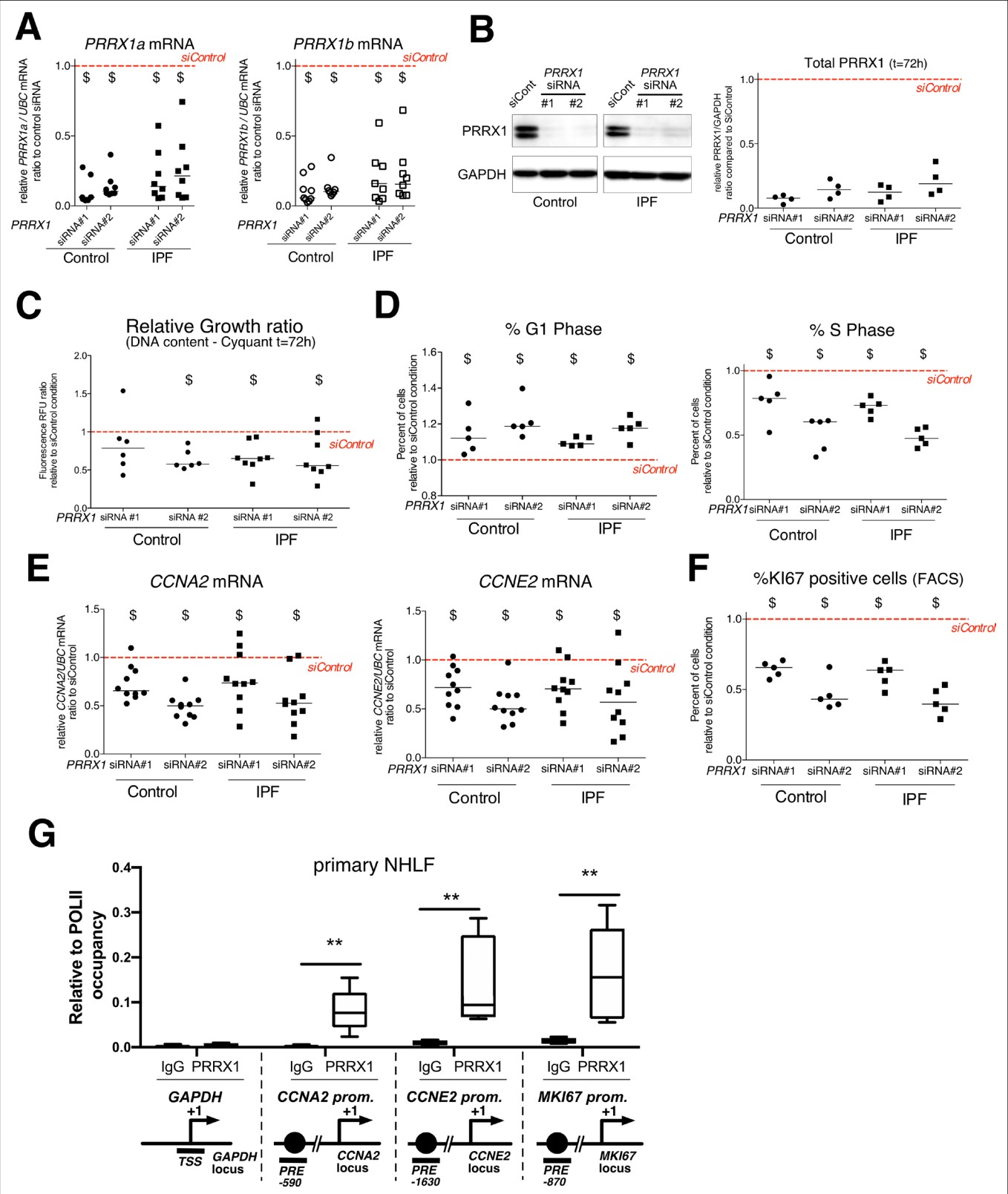

**Figure 4.** *PRRX1* knock down decreased cell proliferation. (**A**) Dot plots with median showing *PRRX1a* (black) and *PRRX1b* (white) mRNA expression relative to the siControl condition (red dashed line) in control (circle) and IPF (square) fibroblasts (n=8 per group) treated for 72 hr with *PRRX1* siRNA (#1 or #2). (**B**) Immunoblot showing PRRX1 expression (n=4) in control and IPF fibroblasts treated 72 hr with *PRRX1* siRNA (#1 or #2) or siControl. The quantification of PRRX1 expression relative to GAPDH (loading control) is displayed as dot plot with median. (**C**) Dot plots with median showing the

*Figure 4 continued on next page*

*Figure 4 continued*

relative growth ratio of control (n=6) and IPF (n=8) fibroblasts stimulated 72 hr with FCS 10% and treated with *PRRX1* siRNA compared to siControl. (**D**) Dot plots with median showing the percent of cells in G1 (right) or S (left) phase in control and IPF fibroblasts (n=5) stimulated 72 hr with FCS and *PRRX1* siRNA relative to siControl. (**E**) Dot plots with median showing mRNA expression of *CCNA2* and *CCNE2* relative to siControl in control and IPF fibroblasts stimulated 72 hr with FCS and treated with *PRRX1* siRNA (n=10). (**F**) Dot plots with median showing the percent of cells positive for Ki67 marker in control and IPF fibroblasts stimulated 72 hr with FCS 10% and treated with *PRRX1* siRNA relative to siControl (n=5). (**G**) ChIP analysis for PRRX1 recruitment at the promoter of *GAPDH, CCNA2, CCNE2,* and *MKI67* in NHLF (n=5) relative to RNA POL-II occupancy, displayed as boxes with median and min to max. The diagrams of the different loci are showing the PRRX1 response element position relative to the TSS. The PCR amplified regions are underscored. *Abbreviations: FCS (fetal calf serum); TSS (transcription starting site); IgG (Immunoglobulin); PRE (PRRX1 responses element); SRE (SRF response element), control siRNA sequence (siControl).* Wilcoxon signed-rank test, $ p≤0.05, Wilcoxon matched-paired signed rank test ** p<0.01.

The online version of this article includes the following source data and figure supplement(s) for figure 4:

**Source data 1.** The zip folder contains the data presented in panels A-G (in an Excel document).Labelled (.pdf) and raw (folder) blot images showed in panel B are also included.

**Figure supplement 1.** PRRX1 inhibition impacted fibroblast proliferation.

**Figure supplement 1—source data 1.** The zip folder contains the data presented in panels B and D (in an Excel document).

In conclusion, our results strongly suggested that PRRX1 TFs are involved in the regulation of fibroblast proliferation in vitro.

## PRRX1 TFs are required for the induction of alpha smooth muscle actin during TGF-β1-driven myofibroblastic differentiation

Next, we investigated the effects of PRRX1 TFs partial loss of function on myofibroblastic differentiation in primary control and IPF lung fibroblasts. In an appropriate microenvironment, fibroblasts can differentiate by acquiring contractile properties such as expression of alpha smooth muscle actin (ACTA2); gamma smooth muscle actin (*ACTG2*) and Transgelin (*TAGLN / SM22*) and becoming active producers of extracellular matrix (ECM) proteins (such as Collagen 1, COL1; Fibronectin, FN1; Tenascin C, TNC and Elastin, ELN). Aberrant activation of fibroblasts into myofibroblasts is thought to be a major driver of lung fibrogenesis (*Fernandez and Eickelberg, 2012*; *Martinez et al., 2017*).

We evaluated the effects of PRRX1 modulation on the expression of myofibroblast markers such as ACTA2, COL1 and FN1 at basal condition. PRRX1 TF loss of functions did not robustly modify the basal expression of these markers at the mRNA and proteins levels after 48 hr of treatment as assayed respectively by qPCR and western blot (*Figure 5—figure supplement 1*). PRRX1 has been also implicated in a positive feed-back loop in which TWIST1 directly increased PRRX1 which subsequently induced Tenascin-C that itself stimulated TWIST1 activity in Cancer associated fibroblast (CAF), and in dermal and fetal Human lung fibroblast lines (*Yeo et al., 2018*). However, we observed that *PRRX1* inhibition with siRNA failed to modulate *TNC* and *TWIST1* mRNA levels in adult primary control and IPF lung fibroblasts at basal condition (*Figure 5—figure supplement 1*).

As mentioned before, TGF-β1 is a major regulator of myofibroblastic differentiation. We determined whether PRRX1 TFs may regulate myofibroblastic differentiation upon TGF-β1 stimulation. Control and IPF primary lung fibroblasts were first treated with *PRRX1* siRNA for 48 hr and then stimulated with 1 ng/ml TGF-β1 for 48 hr. The inhibition of *PRRX1* TFs impacted the upregulation of contractile-associated actin isoforms at the mRNA levels such as *ACTA2* (α-SMA) and *ACTG2* (γ-SMA) in response to TGF-β1 stimulation (*Figure 5A* and *Figure 5—figure supplement 2*) while the expression of the actin binding protein *TAGLN* (SM22) was not perturbed (*Figure 5—figure supplement 2*). The effect of PRRX1 inhibition upon ACTA2 upregulation was confirmed at the protein level in both control and IPF fibroblasts (*Figure 5B*).

With respect to ECM synthesis, *PRRX1* knock down did not influence *FN1* or *COL1A1* upregulation after TGF-β1 stimulation, both at mRNA and protein levels (*Figure 5—figure supplement 2*) in lung fibroblasts. Nevertheless, other ECM proteins associated with IPF were modulated after PRRX1 down regulation in presence of TGF-β1. For instance, the expression of *TNC* mRNA was increased in *PRRX1* siRNA-treated control and IPF lung fibroblasts compared to control siRNA treated in presence of TGF-β1 (*Figure 5—figure supplement 2*). Meanwhile, the expression of *ELN* mRNA was downregulated in IPF lung fibroblasts (*Figure 5—figure supplement 2*) after TGF-β1 stimulation.

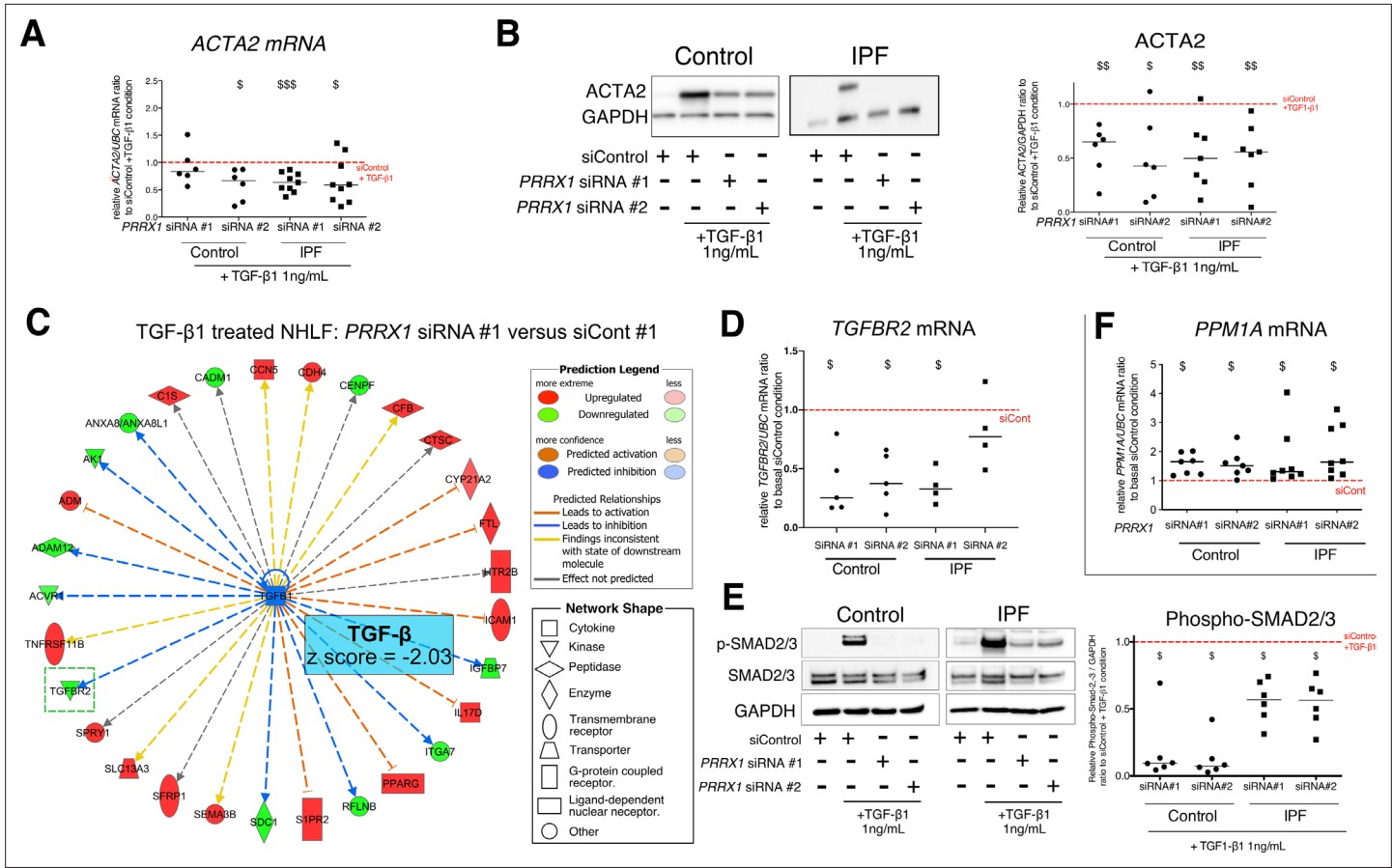

**Figure 5.** *PRRX1* inhibition decreased myofibroblast differentiation upon TGF-β1 stimulation. (**A**) Dot plots with median showing the mRNA expression of *ACTA2* relative to the siControl +TGF-β1 condition (red dashed line), in control (circle, n=6) and IPF (square, n=9) lung fibroblasts treated with TGF-β1 and PRRX1 siRNA (#1 or #2). (**B**) Immunoblot showing ACTA2 expression in control (n=6) and IPF (n=7) fibroblasts treated with siControl in absence or presence of TGF-β1 or with *PRRX1* siRNA and TGF-β1. The quantification of ACTA2 expression relative to GAPDH (loading control) in control and IPF fibroblasts treated with control or *PRRX1* siRNA in presence of TGF-β1 relative to siControl +TGF-β1 condition is displayed as dot plot with median on the right. (**C**) Ingenuity Pathway Analysis of whole transcriptome in NHLF treated for 48 hr with *PRRX1* siRNA in the presence of TGF-β1 indicated that the best predicted upstream regulator was TGFB1 (z score = −2.03, *PRRX1* siRNA#1 versus siControl#1, n=2). Inhibition of *TGFBR2* is framed with a green dashed border. Figure legend displays molecules and function symbol types and colors. (**D**) Dot plots with median showing the mRNA expression of *TGFBR2* (n=4 control HLF and n=5 IPF HLF) relative to siControl in control and IPF fibroblasts treated for 48 hr with *PRRX1* siRNA. (**E**) Immunoblot showing phospho-SMAD2/3 and SMAD2/3 expression in control and IPF fibroblasts treated for 30 min with TGF-β1 after 48 hr transfection with *PRRX1* siRNA. The quantification of phospho-SMAD2/3 and SMAD2/3 expression relative to GAPDH (loading control) in control (n=6) and IPF (n=6) lung fibroblasts treated for 30 min with TGF-β1 after 48 hr transfection with *PRRX1* siRNA relative to siControl +TGF-β1 condition (red dashed line), is displayed as dot plot with median on the right. (**F**) Dot plots with median showing the mRNA expression of *PPM1A* (n=7 control HLF and n=8 IPF HLF) relative to siControl in control and IPF fibroblasts treated for 48 hr with *PRRX1* siRNA. *Abbreviations: control siRNA sequence (siControl)*, β-Tubulin (TUB). Wilcoxon signed-rank test, $ p≤0.05, $$ p≤0.01, $$$ p≤0.001.

The online version of this article includes the following source data and figure supplement(s) for figure 5:

**Source data 1.** The zip folder contains the data presented in panels A-B and D-F (in an Excel document).

**Figure supplement 1.** PRRX1 inhibition did not modulate the basal expression of myofibroblastic markers in control and IPF fibroblasts.

**Figure supplement 1—source data 1.** The zip folder contains the data presented in panels A-C (in an Excel document).

**Figure supplement 2.** Effect of PRRX1 inhibition upon the expression of myofibroblastic markers after TGF-β1 stimulation in control and IPF fibroblasts.

**Figure supplement 2—source data 1.** The zip folder contains the data presented in panels A-D (in an Excel document).

**Figure supplement 3.** *PRRX1* knock-down globally impacted TGF-β pathway response in NHLF.

**Figure supplement 3—source data 1.** The zip folder contains the data presented in panel A (in an Excel document).

**Figure supplement 4.** PRRX1 TFs modulate TGF-β / SMAD signalling cascade by regulating TGFBR2 and PPM1A expressions.

**Figure supplement 4—source data 1.** The zip folder contains the data presented in panels A-E (in an Excel document).

*Figure 5 continued on next page*

*Figure 5 continued*

**Figure supplement 5.** *Prrx1* complete loss of function in mouse embryonic fibroblasts (MEFs) strongly inhibited myofibroblast differentiation upon TGF1-β1 stimulation but promoted lung spheroids formation.

**Figure supplement 5—source data 1.** The zip folder contains the data presented in panels B-D (in an Excel document).

## PRRX1 TFs modulate SMAD2 and SMAD3 phosphorylation in response to TGF-β1 by regulating the expression of TGFβ Receptor 2 (TGFBR2) and the serine/ threonine phosphatase PPM1A

To better appreciate the *PRRX1* siRNA effects upon TGF-β1 pathway, whole transcriptome profiling was performed on NHLF treated with *PRRX1* or control siRNAs for 48 hr and then in presence or absence of TGF-β1 (for an additional 48 hr). Ingenuity Pathway Analysis at 96 hr indicated that the most significantly modulated pathway by *PRRX1* inhibition was the TGF-β1 pathway, which was significantly inhibited in TGF-β1-stimulated NHLF treated with *PRRX1* siRNA compared to control siRNA (*Figure 5C*, *Figure 5—figure supplement 3* and *Supplementary file 2*).

Interestingly, *PRRX1* knockdown significantly affected the expression of the transmembrane Serine/Threonine kinase receptor *TGFBR2*, a key component of the TGF-β pathway (*Figure 5C* and *Figure 5—figure supplement 3*). We confirmed this observation in control and IPF fibroblasts at mRNA and protein levels (*Figure 5D* and *Figure 5—figure supplement 4*). We performed a ChIP assay to investigate a possible interaction of PRRX1 TFs with *TGFBR2* gene promoter regions. However, we detected no enrichment in PRRX1 TF binding in *TGFBR2* promoter regions by ChIP in primary NHLF (data not shown).

TGFBR2 is part of the receptor complex with TGFBR1 controlling TGF-β/SMAD signaling cascade by promoting SMAD2 and SMAD3 phosphorylation upon TGF-β1 stimulation. First, *PRRX1* siRNA did not impact total SMAD2 and SMAD3 expression at the mRNA and protein levels in control and IPF lung fibroblasts (data not shown). The mRNA levels of inhibitory *SMAD7* were also unaffected (data not shown). Next, we assayed SMAD2 and SMAD3 phosphorylation in control and IPF fibroblasts treated with *PRRX1* siRNA, compared to cells transfected with control siRNA, in presence or absence of TGF-β1 (30 min stimulation) (*Figure 5E*). In control and IPF fibroblasts, PRRX1 knock down inhibited TGF-β1-induced SMAD2 and SMAD3 phosphorylation (*Figure 5E*). Most importantly, *PRRX1* knock down did not inhibit the activation of non-canonical/SMAD-independent TGF-β receptor-mediated signaling pathway such as AKT and JNK, in both control and IPF fibroblast (data not shown).

To understand this discrepancy between inhibition of SMAD2/3 phosphorylation and persistent activation of non-canonical pathways, we investigated whether PRRX1 TFs may regulate the expression of intracellular phosphatases known to control SMAD2 and SMAD3 phosphorylation downstream of TGF-β receptor activation (*Bruce and Sapkota, 2012*). We observed that *PRRX1* siRNA-mediated inhibition was associated with an increase of PPM1A, a phosphatase member of the PP2C protein family, at both mRNA and protein levels compared to control and IPF fibroblasts treated with control siRNA (*Figure 5F* and *Figure 5—figure supplement 4*). In addition, siRNA-mediated inhibition of *PPM1A* partially rescued SMAD3 phosphorylation levels after TGF-β1 stimulation in *PRRX1* siRNA-treated control and IPF fibroblasts (*Figure 5—figure supplement 4*). Next, we performed a ChIP assay in primary NHLF to investigate a possible interaction of PRRX1 TFs with *PPM1A* gene *loci*. Indeed, we detected an enrichment in PRRX1 TF binding in *PPM1A* promoter regions (*Figure 5—figure supplement 4*).

Altogether, our results suggested that PRRX1 TFs contribute to myofibroblastic differentiation upon TGF-β1 stimulation in lung fibroblasts. The effect is at least partially mediated through the regulation of TGFβ receptor 2 and PPM1A phosphatase expression. The role of PRRX1 TFs during myofibroblastic differentiation downstream of TGFβ1 was confirmed in *Prrx1*⁻/⁻ mouse embryonic fibroblasts (MEFs) compared to wild-type counterpart (see *Figure 5—figure supplement 5*). In this case, *Prrx1* complete loss of function inhibited both ACTA2 and COL1 upregulation at both mRNA and protein levels after TGF-β1 stimulation at 1 and 10 ng/ml for 48 hr (*Figure 5—figure supplement 5*). We also investigated whether PRRX1 loss could perturb epithelial homeostasis and regeneration using a lung spheroid assay (*Konishi et al., 2022*). Using wild-type and *Prrx1*⁻/⁻ MEFs as stromal cells cultured with wild-type primary mouse lung epithelial cells, we showed that *Prrx1* loss did not dampen but rather promoted epithelial spheroid formation (*Figure 5—figure supplement 5*).

## PRRX1 TFs expression levels are upregulated in the bleomycin-induced model of lung fibrosis

In the light of our in vitro results regarding PRRX1 role in the control of fibroblast proliferation and TGF-β1 responsiveness, we investigated whether alteration in PRRX1 expression may also contribute to fibrogenesis in the bleomycin-induced model of lung fibrosis (single intratracheal instillation *Moshai et al., 2014*). In this model, the expression levels of both *Prrx1* isoforms mRNA were mainly increased during the fibrotic phase from day 7 to 28 compared to the control saline-treated animals (*Figure 6— figure supplement 1*). The upregulation of PRRX1 expression level was confirmed at the protein level only from day 14 to 28 during fibrosis phase peak (*Figure 6—figure supplement 1*). Nevertheless, a slight decreased expression of *Prrx1b* mRNA and PRRX1 total protein levels was observed at day 21 compared to day 14 and 28 during the late fibrosis phase (*Figure 6—figure supplement 1*).

As in control Human lungs, PRRX1-positive cells were detected only within the peri-vascular and peri-bronchiolar spaces in saline control mice, while the distal alveolar space and the bronchiolar epithelium were devoid of PRRX1 staining as assayed by immunohistochemistry. Meanwhile, PRRX1-positive cells were detected in the remodeled fibrotic area of bleomycin-treated animals at day 14 (*Figure 6—figure supplement 1*). To further characterize the specific populations of fibroblasts expressing *Prrx1* in mouse lung, we leveraged the published single-cell transcriptomic analysis from FibroXplorer (*Buechler et al., 2021*) and analyzed the lung dataset at baseline or after bleomycin challenge. In control lungs, *Prrx1* is mainly present in *Pi16*-positive adventitial and *Col15a1* parenchymal 'universal' fibroblast populations (*Buechler et al., 2021*), which also express *Col14a1* (see *Figure 6— figure supplement 1*). Indeed, *Prrx1* was previously described as potential master transcription factor in the mesenchymal lung cell subtypes expressing *Col14a1* (*Xie et al., 2018*). Interestingly, *Prrx1* is also detected in the fibrosis-associated *Lrrc15/Cthrc1*-positive fibroblast population producing the highest levels of ECM proteins (*Tsukui et al., 2020*) in animals treated with bleomycin, as shown in *Figure 6—figure supplement 1*. In summary, our results indicated that PRRX1 TFs upregulation was associated with fibrosis development in the bleomycin-induced model of lung fibrosis.

## In vivo inhibition of PRRX1 dampens experimental lung fibrosis

Since *Prrx1* loss of function is associated with perinatal lethality in *Prrx1*[-/-] pups (*Ihida-Stansbury et al., 2004*; *Lu et al., 1999*; *Martin et al., 1995*), we sought first to evaluate *Prrx1* function during lung fibrosis using *Prrx1*[+/-] +/- mice. We observed that the loss of one *Prrx1* allele was not associated with any haploinsufficiency (*Figure 6—figure supplement 2*) and those *Prrx1*[+/-] +/- mice were not protected from lung fibrosis at day 14 after intratracheal instillation of bleomycin (*Figure 6—figure supplement 2*).

To evaluate the involvement of PRRX1 TFs in pulmonary fibrosis in vivo, we then chose to treat wild type mice with a third generation antisense LNA-modified oligonucleotide (ASO) targeting both *Prrx1* isoforms by an endotracheal route to target specifically the lung. After treating naive mice with PBS, control, or *Prrx1* ASO every other day for a week, there was no evidence of inflammation or upregulation of fibrosis markers associated with *Prrx1* inhibition in healthy lungs when compared to the PBS and control ASO groups (data not shown). In the bleomycin-induced model of lung fibrosis, the control or *Prrx1* ASO were administrated during the fibrotic phase (from day 7–13 - see *Figure 6A*). As compared to control ASO, *Prrx1* ASO strongly reduced the expression of both *Prrx1* isoforms at the mRNA and protein levels (*Figure 6A–B*) and reduced the extent of lung lesions on day 14 (*Figure 6C*). Lung collagen content was decreased as assessed with picrosirius staining, immunohistochemistry and hydroxyproline assay (*Figure 6D–F*). A similar decrease in ACTA2 staining was observed in *Prrx1* ASO-treated animals at day 14 by immunohistochemistry (*Figure 6E* and *Figure 6—figure supplement 3*). *Prrx1* ASO treatment also dampened the accumulation of PDGFR-positive mesenchymal cells in bleomycin-treated animals compared to the control ASO bleomycin group at day 14 by immunohisto-chemistry (*Figure 6—figure supplement 3*).

In addition, *Prrx1* ASO decreased *Col1a1, Fn1* and *Acta2* mRNA content (*Figure 7A*) in PRRX1 ASO-treated bleomycin mice compared to control ASO-treated ones. Finally, the expression levels of COL1, FN1 and ACTA2 were also decreased at the protein level as assayed by western blot (*Figure 7B–C*). The dampened fibrosis development observed in PRRX1 ASO-treated bleomycin mice was also associated with a decrease in key fibrosis mediators such as *Tgfb1* and *Ctgf* as well as inflammatory markers as *Tnf* and *Serpin-1* at the mRNA level (*Figure 6—figure supplement 3*). As

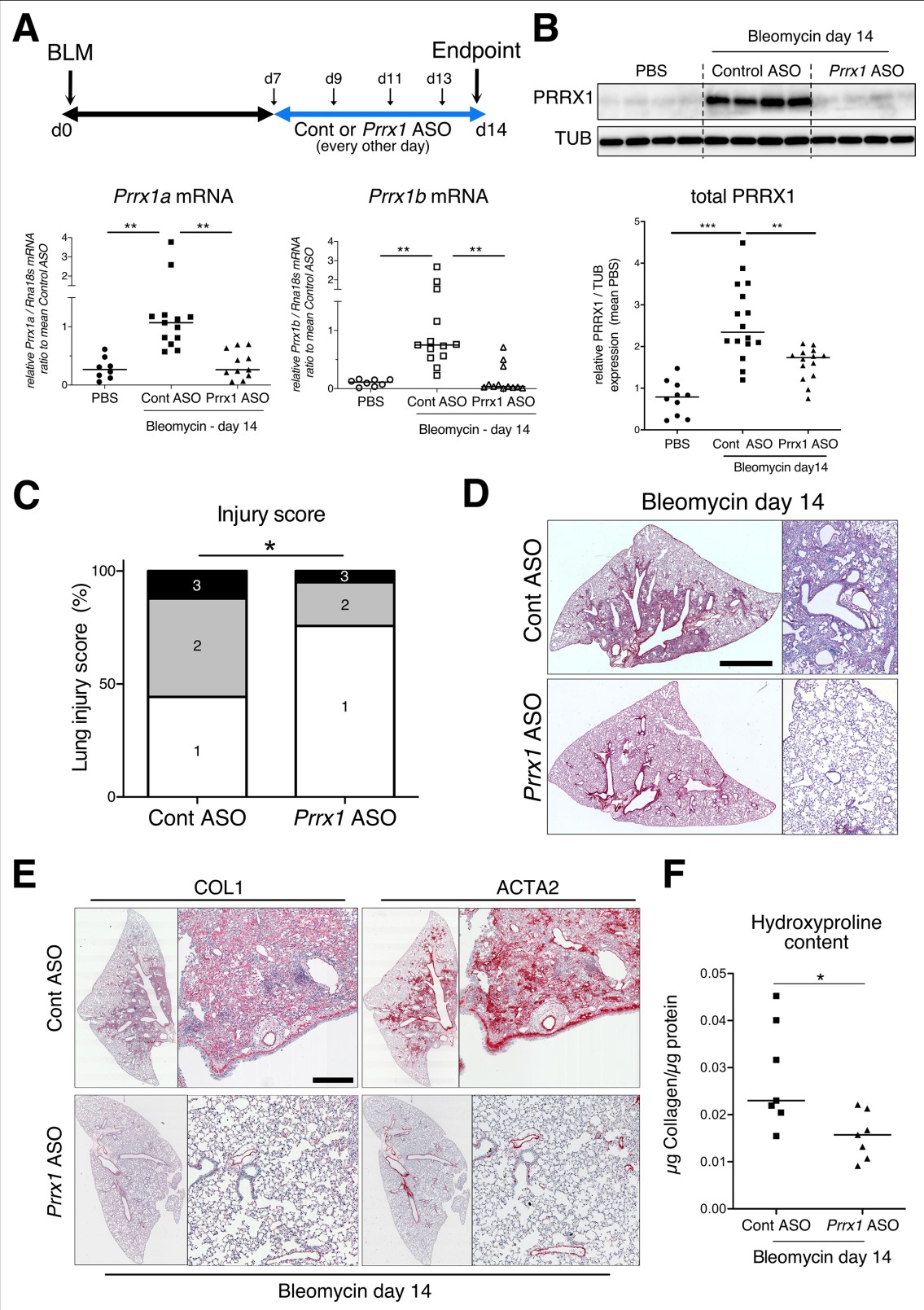

**Figure 6.** PRRX1 inhibition attenuates lung fibrosis in bleomycin murine model. (**A**) Timeline of bleomycin and ASO treatments. Mice were first treated with bleomycin at day zero and then with control or *Prrx1* ASO every other day from day 7, during the active fibrotic phase (in blue). Lungs were harvested at day 14. Lower part: dot plots with median showing the mRNA expression of *Prrx1a* (black) and *Prrx1b* (white) isoforms at day 14 (n=8) in PBS (circle) mice and bleomycin mice treated with control ASO (square, n=13) or *PRRX1* ASO (triangle, n=11). (**B**) Immunoblot showing PRRX1

*Figure 6 continued on next page*

*Figure 6 continued*

expression at day 14 in PBS mice and bleomycin mice treated with Control ASO or *Prrx1* ASO. TUB was used as loading control. The quantification of PRRX1 expression relative to TUB at day 14 in PBS mice (circle, n=9) and bleomycin mice treated with Control ASO (square, n=15) or *Prrx1* ASO (triangle, n=14) is displayed as dot plot with median on the lower panel. (**C**) Injury score at day 14 of bleomycin mice treated with *Prrx1* ASO or Control ASO. (**D**) Representative images (n=7 per group) showing picrosirius staining (red) at day 14 in bleomycin mice treated with Control ASO or *Prrx1* ASO. (**E**) Representative immunohistochemistry images (n=7 per group) showing COL1 (left panel) and ACTA2 (right panel) staining (red) at day 14 in bleomycin mice treated with Control ASO or *Prrx1* ASO. Nuclei were counterstained with hematoxylin. (**F**) Dot plot with median showing the relative Collagen content as measured by hydroxyproline at day 14 in bleomycin mice treated with control ASO (square, n=7) or *PRRX1* ASO (triangle, n=7). Scale bar: 80 µm in low-magnification images and 40 µm in high-magnification ones. *Abbreviations: d (day), Control (Cont), Antisense oligonucleotide (ASO).* Kruskal-Wallis test with Dunns post-test (**A and B**), Fisher's exact test (**C**) and Mann Whitney U test (**F**); *p≤0.05, **p≤0.01, ***p≤0.001.

The online version of this article includes the following source data and figure supplement(s) for figure 6:

**Source data 1.** The zip folder contains the data presented in panels A-C and F (in an Excel document).

**Figure supplement 1.** PRRX1 is increased during fibrotic phase in mice bleomycin-induced fibrosis.

**Figure supplement 1—source data 1.** The zip folder contains the data presented in panels A-B (in an Excel document).

**Figure supplement 2.** lack of haploinsufficiency and lung fibrosis reduction in *Prrx1*+/- +/- mice.

**Figure supplement 2—source data 1.** The zip folder contains the data presented in panels A-B and E-G (in an Excel document).

**Figure supplement 3.** PRRX1 inhibition decreases the accumulation of myofibroblasts as well as the expression of fibrosis and inflammatory markers in bleomycin mice.

**Figure supplement 3—source data 1.** The zip folder contains the data presented in panels A-E (in an Excel document).

mentioned previously, *Prrx1* was recently identified as the master transcription factor in the *Col14a1* subtype mesenchymal cell during fibrogenesis in this experimental model (*Xie et al., 2018*). Interestingly, the expression of *Col14a1* mRNA was also strongly decreased in the *Prrx1* ASO-treated animals compared to control ASO at day 14 as assayed by qPCR (*Figure 6—figure supplement 3*). With respect to another fibrosis-associated key ECM protein, *Eln* and *Tnc* mRNA levels were also decreased in the *Prrx1* ASO-treated bleomycin group (*Figure 6—figure supplement 3*).

The expression levels of the proliferation marker *Mki67* was decreased at the mRNA levels in the *Prrx1* ASO-treated animals compared to control ASO at day 14 as assayed by qPCR (*Figure 7D*). An overall decrease in KI67-positive cells was also observed by immunochemistry in the *Prrx1* ASO-treated animal lungs after bleomycin challenge compared to control ASO ones at day 14 (*Figure 7D*). Such decrease in KI67 positivity was also confirmed in mesenchymal PDGFR-positive cells after performing PDGFR KI67 double staining in the *Prrx1* ASO-treated animals compared to control ASO at day 14 (*Figure 6—figure supplement 3*).

We also investigated the potential effect of *Prrx1* inhibition during late fibrosis phase after bleomycin treatment at day 28 by treating the animals every other day with either control or *Prrx1* ASO between from day 21 to 27 (*Figure 7—figure supplement 1*). In this case, the effects of *Prrx1* inhibition during late fibrosis phase were less potent as measured by hydroxyproline content, even though the injury score, accumulation of PDGFR-positive cells by immunohistochemistry, protein levels of COL1 and FN1 by immunoblots were still decreased in *Prrx1* ASO-treated animals compared to control ASO after bleomycin insult at day 28 (see *Figure 7—figure supplement 1*).

Overall, these data demonstrate that PRRX1 targeting in the lung has the potential to inhibit lung fibrosis development.

## PRRX1 inhibition attenuates lung fibrosis in mouse and human precision-cut lung slices (PCLS)

To confirm the effect of the *Prrx1* ASO in a second model of lung fibrosis, we took advantage of a well-established ex vivo model of lung fibrosis using precision-cut lung slices (PCLS) derived from mouse and Human lung samples (*Lehmann et al., 2018*). PCLS have the major advantage to include the lung primary cell populations in a three-dimensional preserved lung architecture and microenvironment. Our *Prrx1* ASO was designed to target both Human *PRRX1* and mouse *Prrx1* TFs orthologs. In basal condition, *Prrx1* knock down was associated with a decrease in *Acta2* mRNA expression in mouse PCLS (*Figure 8—figure supplement 1*). No effect on *ACTA2*, *COL1A1*, and *FN1* mRNA was observed in control Human PCLS treated with *Prrx1* ASO at baseline (*Figure 8—figure supplement 1*). However, we observed a decreased expression of *ACTA2*, *COL1A1* and *FN1* mRNA in three IPF

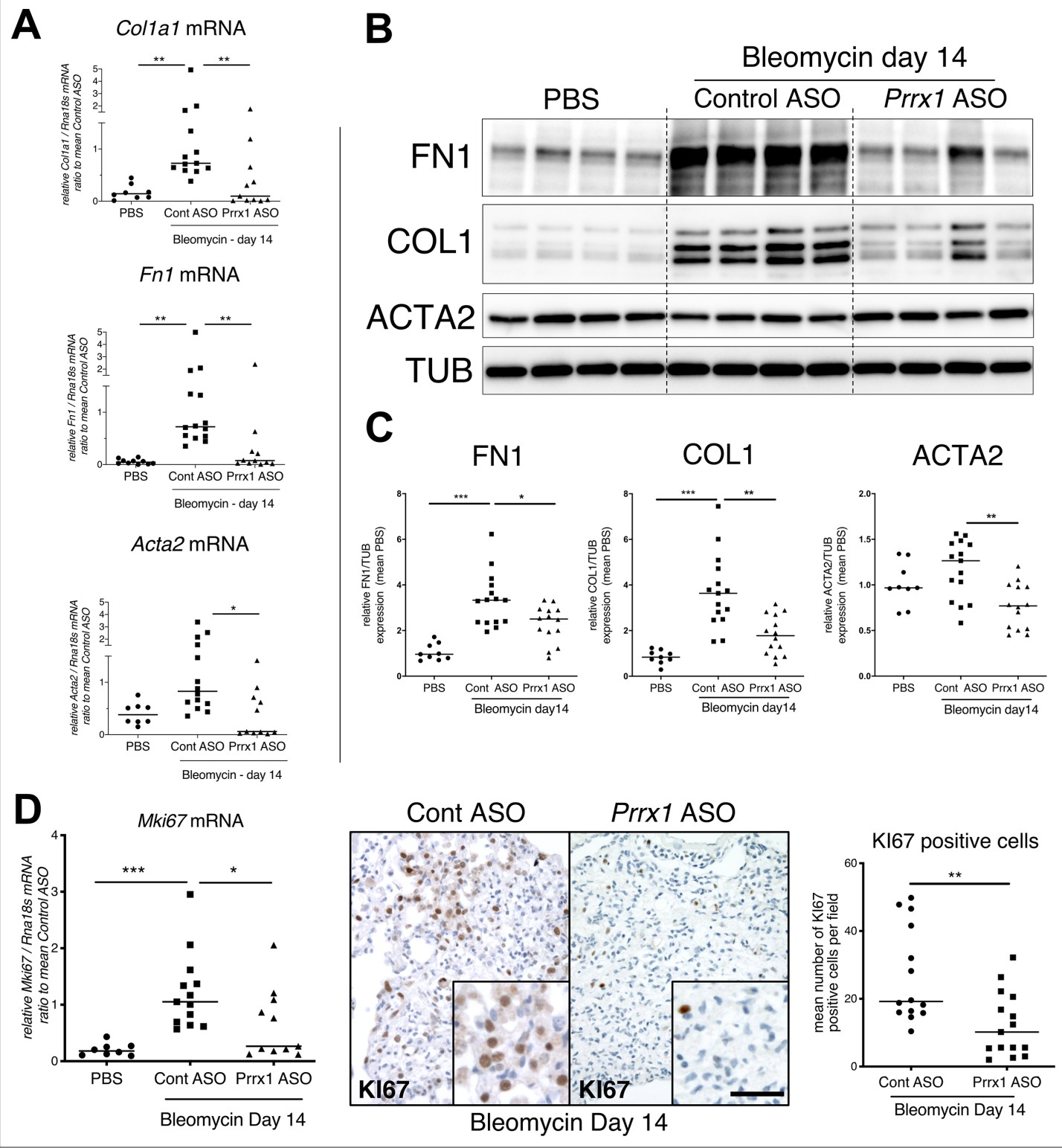

**Figure 7.** PRRX1 inhibition decreases fibrosis markers in bleomycin mice. (**A**) Dot plots with median showing the mRNA expression of *Col1a1*, *Fn1* and *Acta2* at day 14 in PBS mice (circle, n=8) and bleomycin mice treated with Control ASO (square, n=13) or *Prrx1* ASO (triangle, n=11). (**B**) Immunoblot showing FN1, COL1, and ACTA2 expression at day 14 in PBS mice and bleomycin mice treated with Control ASO or *Prrx1* ASO. TUB was used as loading control. (**C**) Quantification of FN1, COL1, and ACTA2 relative expression to TUB at day 14 in PBS mice (n=9) and bleomycin mice treated with Control ASO (n=15) or *Prrx1* ASO (n=14) (**D**) Left panel: dot plots with median showing *Mki67* mRNA expression of at day 14 in PBS mice (circle, n=8) and bleomycin mice treated with Control ASO (square, n=13) or *Prrx1* ASO (triangle, n=11). Middle panel: representative immunohistochemistry pictures

*Figure 7 continued on next page*

*Figure 7 continued*

(n=14 per group) showing KI67 staining (brown) in bleomycin treated with Control ASO (left) or *Prrx1* ASO (right) mice at day 14. The quantification of the number of KI67-positive cells per high magnification field is shown on the right as dot plots with median. Scale bar: 40 µm in low-magnification images and 20 µm in high-magnification ones. *Abbreviations: Control (Cont), Antisense oligonucleotide (ASO).* Kruskal-Wallis test with Dunns post-test (**A, B**) and Mann Whitney U test (**C**); *p≤0.05, **p≤0.01 f, ***p≤0.001.

The online version of this article includes the following source data and figure supplement(s) for figure 7:

**Source data 1.** The zip folder contains the data presented in panels A-D (in an Excel document).

**Figure supplement 1.** PRRX1 inhibition during late fibrosis phase reduces several fibrosis features and markers.

**Figure supplement 1—source data 1.** The zip folder contains the data presented in panels C-E and G (in an Excel document).

PCLS treated with *PRRX1* ASO compared to control ASO-treated ones (**Figure 8—figure supplement 1**).

Next, mouse or Human control lung PCLS were treated with a fibrosis cytokine cocktail (FC) consisting of TGF-β1, PDGF-AB, TNFα, and LPA to trigger fibrosis-like changes (**Lehmann et al., 2018**). In both mouse and Human PCLS, *ACTA2*, *COL1A1,* and *FN1* mRNA upregulation was lessened in FC-stimulated PCLS with *PRRX1* ASO compared to control (**Figure 8**). We confirmed those findings at the protein levels for ACTA2 and COL1 by western blot in mouse PCLS (**Figure 8**). In Human PCLS stimulated with FC, morphological analysis revealed that *PRRX1* ASO treatment was associated with a decreased Collagen accumulation compared to control ASO (**Figure 8**).

Altogether, these results demonstrate that inhibition of PRRX1 transcription factors, using an ASO approach, reduced fibrosis development in vivo and ex vivo.

## Discussion

This is the first study to evidence the critical role of the PRRX1 transcription factors in lung fibrosis pathophysiology. Our results demonstrate that (1) PRRX1 TFs are upregulated in mesenchymal cells accumulating in the fibrotic areas of IPF lungs, (2) the expression of PRRX1 TFs is positively regulated by cues associated with an undifferentiated phenotype in control and IPF primary lung fibroblasts, (3) PRRX1 TFs are required for proliferation as well as proper myofibroblastic differentiation in vitro (see **Figure 9** for summary). We identified the underlying mechanisms, including PRRX1 TFs effects on cell cycle (modulation of cyclins and MKI67) and on SMAD 2/3 phosphorylation (regulation of TGFBR2 and phosphatase PPM1A) respectively (**Figure 9**). Finally, inhibition of *Prrx1* with LNA-modified ASO strongly impacted lung fibrosis development in in vivo and ex vivo preclinical models.

### PRRX1 expression is restricted to certain fibroblast subtypes in healthy and fibrotic lungs

Our data clearly showed that PRRX1 expression level was upregulated in IPF and restricted to mesenchymal lineages in both control and IPF lung samples as assayed by immunochemistry and confirmed by datamining in single cell transcriptomic studies (**Adams et al., 2020**; **Reyfman et al., 2019**). This upregulation of PRRX1 protein levels in IPF lungs may likely reflect the accumulation of PRRX1-positive cells during pulmonary fibrogenesis.

Analysis of the IPF cell atlas sc-RNAseq dataset provided insights into the expression pattern of PRRX1 mRNA in various fibroblast/mesenchymal lung cell subtypes in normal and fibrotic lungs. In both human and mouse lungs, *PRRX1/Prrx1* TFs were mainly expressed by resident adventitial and parenchymal fibroblasts, which are considered as universal fibroblast populations that may act as potential progenitors for specialized fibroblast subtypes (**Buechler et al., 2021**). In contrast, *Prrx1* expression was not enriched in lung specialized fibroblast subtypes in mouse lung at baseline (*Npnt^pos* alveolar and *Hhip^pos* peri-bronchial clusters in **Buechler et al., 2021**). These findings suggest that PRRX1 expression may be associated with potential mesenchymal progenitors in healthy lungs. Notably, *PRRX1/Prrx1* expression was maintained in fibroblast subtypes enriched during fibrosis, including the *Lrrc5^pos*/ *Cthrc1^pos* population in the mouse bleomycin model, *ACTA2^pos* myofibroblasts, *PLIN2^pos* lipofibroblast-like cells, and *HAS1^hi* fibroblasts in IPF. Previous studies have showed that PRRX1 TFs were involved in adipogenesis (**Du et al., 2013**) as well as potentially influencing the cell fate switch between alveolar myofibroblasts and lipofibroblasts during fetal lung mesodermal development (**Li et al., 2016**). Given

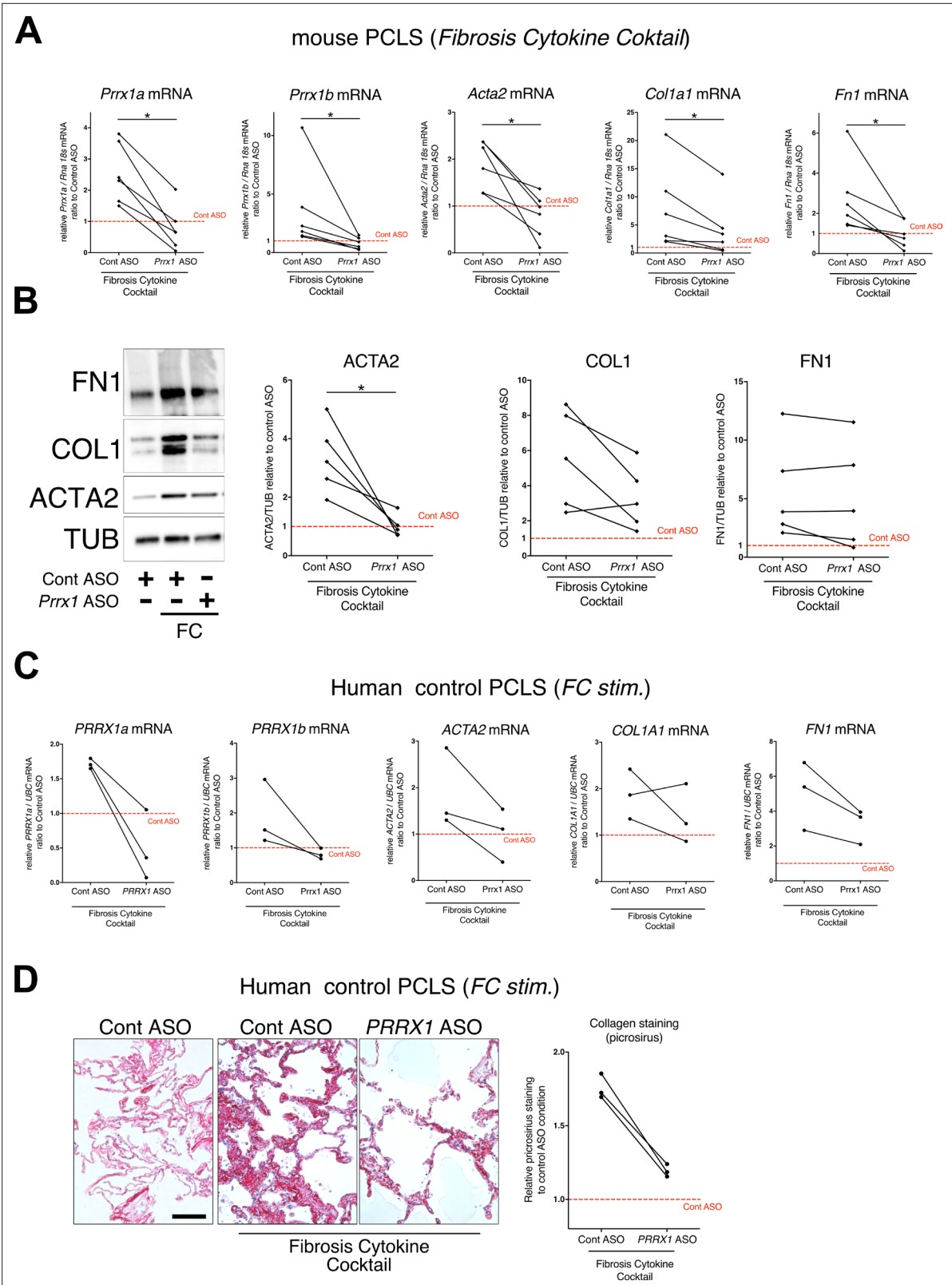

**Figure 8.** *PRRX1* ASO attenuates lung fibrosis in mouse and Human Precision-cut Lung slices (PCLS). (**A**) Before-after plots showing the mRNA expression of *Prrx1a, Prrx1b, Acta2, Col1a1,* and *Fn1* (n=6) relative Control ASO alone condition (red dashed line) in mouse PCLS stimulated with fibrosis cytokine cocktail (FC) and then treated either with control ASO or *PRRX1* ASO. (**B**) Representative immunoblot showing FN1, COL1, and ACTA2 expression (n=5) relative to control ASO alone condition (red dashed line) in mouse PCLS stimulated with FC and then treated either with control or

*Figure 8 continued on next page*

*Figure 8 continued*

*Prrx1* ASO. The corresponding quantifications of ACTA2, COL1, and FN1 expression ratio to Tubulin are displayed as before-after plots on the right. Note that COL1 expression was decreased in 4 out 5 experiments. (**C**) Before-after plots showing the mRNA expression of *PRRX1a, PRRX1b, ACTA2, COL1A1* and *FN1* (n=3) relative Control ASO condition (red dashed line) in Human PCLS treated either with control ASO or *PRRX1* ASO in presence or absence of FC. *COL1A1* upregulation was lessened in 2 out 3 experiments while ACTA2 and FN1 levels were decreased in 3 out of 3 experiments. (**D**) Representative picrosirius staining (n=3) in Human PCLS treated with control ASO alone (left panel, basal condition) or after stimulation with Fibrosis Cytokine cocktail and treated with either control (middle panel) or *PRRX1* (right panel) ASO. Nuclei were counterstained with hematoxylin. The quantification of picrosirius staining relative to control ASO alone (red dashed line) is showed on the right (Before-after plot). Scale bar: 50 μm. *Abbreviations: Precision-Cut Lung slices (PCLS), Fibrosis Cytokine Cocktail (FC), Control (Cont), Antisense oligonucleotide (ASO), Stimulation (stim.).* Wilcoxon test * $P \leq 0.05$.

The online version of this article includes the following source data and figure supplement(s) for figure 8:

**Source data 1.** The zip folder contains the data presented in panels A-D (in an Excel document).

**Figure supplement 1.** effect of *PRRX1* ASO on the expression of myofibroblastic differentiation markers in mouse and Human PCLS at basal condition.

**Figure supplement 1—source data 1.** The zip folder contains the data presented in panels A-D (in an Excel document).

the important role of lipofibroblasts in lung repair and epithelial regeneration (*El Agha et al., 2017*), further investigation is required to fully understand the significance of the observed increase in PRRX1 expression in this fibroblast subtype. Future studies could also explore the effects of targeting *Prrx1* loss specifically in these populations, including the pro-fibrotic *Lrrc5*[pos]/ *Cthrc1*[pos] cluster, to provide a more comprehensive understanding of the role of PRRX1 in lung fibrosis.

### *PRRX1* TFs expression is increased in IPF and regulated by cues promoting an undifferentiated state in primary Human lung fibroblasts

Our in vitro study showed that PRRX1 expression may be controlled by a PGE2/TGF-β balance in lung fibroblasts in vitro. On one hand, PGE2 up-regulated *PRRX1a* and *1b* expression in both control and IPF fibroblasts. Substrate stiffness in physiological range also increased *PRRX1* isoforms expression in a PTGS2 dependent manner. The PTGS2/PGE2 axis is known to promote an undifferentiated state in fibroblasts (*Liu et al., 2010*). On the other hand, signals triggering myofibroblastic differentiation (TGF-β1 stimulation and stiff substrate) decreased *PRRX1* TFs levels in primary lung fibroblasts (*Figure 9*).

Interestingly, several studies reported that PRRX1 expression level was rather increased upon the activation of the TGF-β pathway in other cell types such as mouse embryonic lung mesenchymal cells (*Li et al., 2016*), embryonic mouse 3T3-L1 adipocyte precursor (*Du et al., 2013*) and transformed epithelial cells undergoing EMT (*Ocaña et al., 2012*). PRRX1 upregulation in response to TGF-β1 in the two later cell types promoted their dedifferentiation toward a more plastic phenotype (*Du et al., 2013*; *Ocaña et al., 2012*). Conversely, *PRRX1* downregulation in primary lung fibroblasts grown in presence of TGF-β1 was associated with a differentiation process toward myofibroblastic phenotype. Conditional *Prrx1* loss of function in *Sm22* (*Tagln*) positive CAFs in a mouse model of pancreatic cancer also promoted in vitro an activated phenotype with an increase in ACTA2 and COL1 expression levels (*Feldmann et al., 2021*).

Overall, these different studies and our results strongly suggested that PRRX1 expression is associated with an undifferentiated phenotype. In addition, *PRRX1a* and *1b* TF mRNA expression levels were upregulated only in control fibroblasts seeded on IPF fibroblast-derived 3D ECM in a PDGFR dependent manner. These results suggested that components present in IPF ECM directly modulated the expression of PRRX1 TFs in control fibroblasts independently of ECM stiffness in this assay. Further research will be necessary to determine the mechanisms that activate PGDFR by IPF-fibroblast derived ECM, leading to upregulation of PRRX1 TFs in control fibroblasts.

In conclusion, *PRRX1* TF mRNA levels seemed to be regulated by both ECM origin and stiffness in control fibroblasts, while it was only modulated by the latter in IPF fibroblasts (*Figure 9*).

### PRRX1 TFs drive key basic fibroblast functions involved in fibrogenesis

In adult lung fibroblasts, PRRX1 TFs appeared to strongly influence cell cycle progression and myofibroblastic differentiation, two entangled cellular processes (*Figure 9*). There was generally no difference between control and IPF fibroblasts regarding *PRRX1* functions in those cells (with respect to

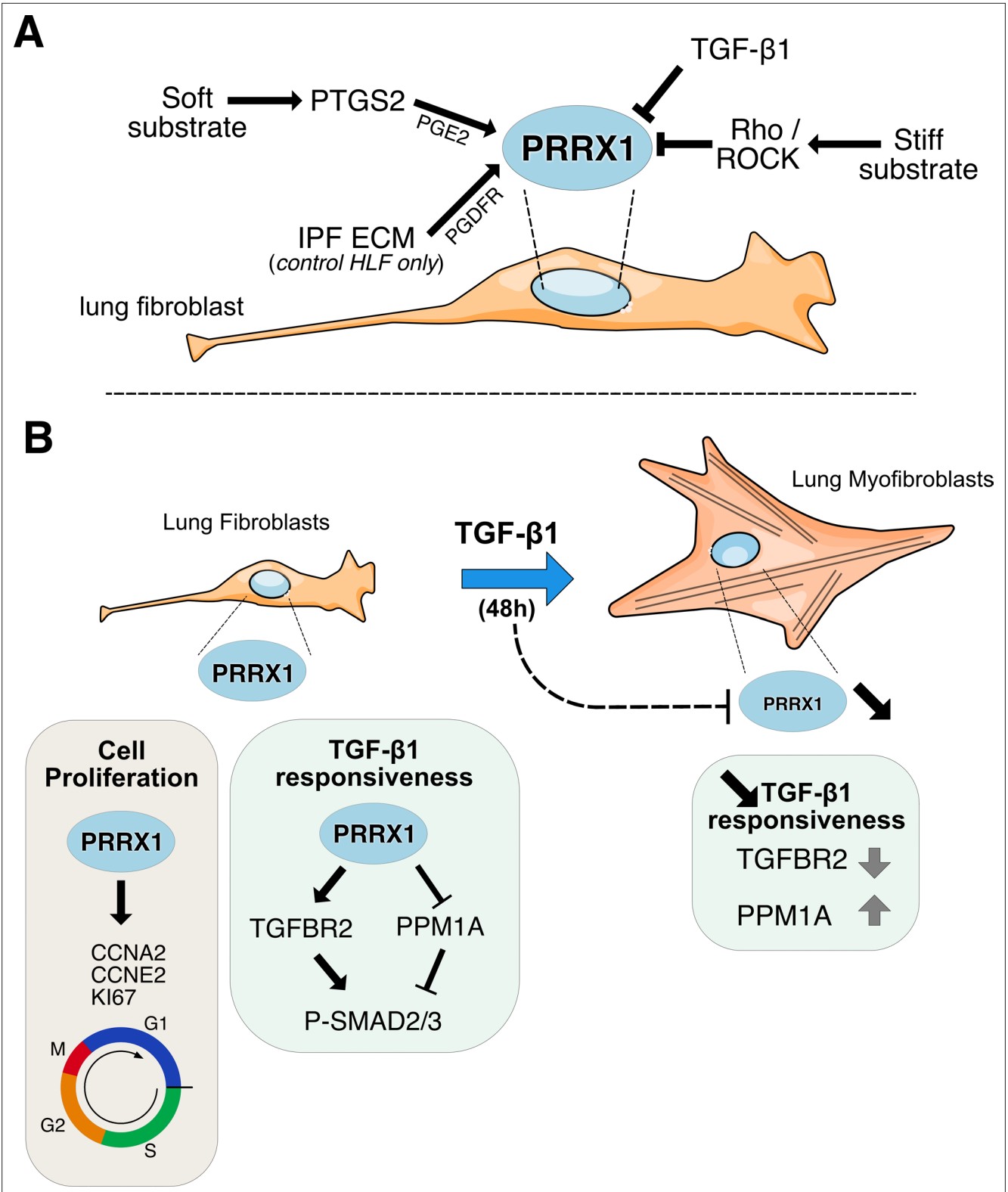

**Figure 9.** Summary sketch of PRRX1 regulation and functions in lung fibroblasts. (**A**) Regulation of *PRRX1* TF expression in lung fibroblasts. On one hand, PRRX1 expression was up-regulated by the anti-fibrotic factor PGE2 and soft culture substrate (in a PTGS2 dependent-manner). IPF fibroblast-derived matrix also increased *PRRX1* TFs expression in a PDGFR dependent manner in control primary lung fibroblasts only. On the other hand, stiff culture substrate (in a Rho/ROCK dependent manner) and TGF-β1 stimulation, which both promote myofibroblastic differentiation, decreased PRRX1 TF expression levels in both control and IPF fibroblasts seeded on plastic. (**B**) Model of PRRX1 function in lung fibroblasts at steady state and after TGF-β1

*Figure 9 continued on next page*

*Figure 9 continued*

driven myofibroblastic differentiation. Left panel: in complete growth medium, PRRX1 TFs influence cell cycle progression by regulating key factors associated with cycle progression during the G1 and S phases (KI67, Cyclin A2 and E2). PRRX1 was detected in the promoter regions of those genes by chromatin immunoprecipitation (ChIP). In presence of PRRX1, the expression of the serine / threonine phosphatase PPM1A is downregulating (PRRX1 TFs binding to PPM1A promoter region was demonstrated by ChIP) and TGFBR2 expression is also maintained. Thus, TGF-β1 responsiveness and the phosphorylation of SMAD2 and SMAD3 may be therefore promoted by PRRX1 TFs in lung fibroblasts at steady state. Right panel: TGF-β1 stimulation of lung fibroblasts will trigger their differentiation into myofibroblasts, concomitantly with a downregulation of PRRX1 TFs at late time point (48 hr). This negative feedback loop could limit cell-responsiveness to long exposure of TGF-β1 by upregulating the expression of PPM1A and downregulating TGFBR2 levels. *Abbreviations: IPF (Idiopathic Pulmonary Fibrosis), HLF (Human Lung Fibroblasts), ECM (Extracellular matrix), G1 (Gap 1 phase 1), S (Synthesis / Replicative phase), G2 (Gap phase 2), M (Mitosis), CCNA2 (Cyclin A2), CCNE2 (Cyclin E2).*

proliferation and myofibroblastic differentiation at least). Overall, this may suggest that *PRRX1* TFs function might be central to fibroblast biology independently of their origin (control versus IPF lungs). However, differential *PRRX1* regulation between control and IPF fibroblasts by the micro-environment or soluble factors could therefore have a higher impact on PRRX1 overall function in lung fibroblasts.

PRRX1 partial loss of function perturbed some key features of myofibroblastic differentiation in response to TGF-β1 stimulation (see *Figure 9*). The expression of markers involved in the acquisition of contractile properties (ACTA2 and *ACTG2*) was decreased in a SMAD2/3-dependent way. The effect of PRRX1 inhibition upon P-SMAD3 was partially mediated through TGFBR2 downregulation and the upregulation of the PPM1A phosphatase (*Bruce and Sapkota, 2012*), which are critical components of the canonical TGF-β/SMAD signaling cascade. Whole transcriptome profiling data performed in NHLF were also consistent with a global impact of *PRRX1* downregulation on TGF-β response in lung fibroblasts.

Meanwhile, the expression levels of key ECM proteins such as Collagen 1 and Fibronectin were still upregulated in TGF-β1 stimulated lung fibroblasts, transfected with *PRRX1* siRNA, whereas *TNC* and *ELN* mRNA expression levels were perturbed. These findings may suggest that broad phenotypical changes are associated with *Prrx1* knockdown in lung fibroblasts. Even though, SMAD3 phosphorylation was impacted, the non-canonical ERK, AKT, and JNK pathways were still fully activated in those stimulated cells. The activation of those pathways has been previously showed to be sufficient to upregulate the expression of FN1 and COL1 in fibroblasts (*Bruce and Sapkota, 2012*; *Fernandez and Eickelberg, 2012*). The effect of PRRX1 TFs on ECM protein expression could be also influenced by the extent of *Prrx1* downregulation since a complete loss of function in *Prrx1⁻ᐟ⁻* MEFs also inhibited the upregulation of COL1 in response to TGF-β1.

Albeit we showed that PRRX1 TFs contribute to myofibroblastic differentiation, a time course analysis revealed that the expression of both PRRX1 isoforms was significantly decreased only at the 48 hr time point in both control and IPF fibroblasts. This late downregulation of PRRX1 in response to TGF-β1, could be the signature of a negative feedback loop to limit cell-responsiveness to TGF-β1 when lung fibroblasts are fully differentiated into myofibroblasts at 48 hr (*Figure 9*). TGF-β1 induced PRRX1 inhibition in lung fibroblasts could also correlate with progressive proliferation loss during the differentiation process.

Overall, we propose that PRRX1 TFs in the lung would maintain mesenchymal cells in a proliferative state but would also act as enablers to promote full myofibroblastic differentiation in response to profibrotic cues such as TGF-β1.

## Inhibition of the mesenchymal PRRX1 transcription factor is sufficient to dampen lung fibrosis in vivo

To inhibit PRRX1 TFs in the lung, we used a LNA-modified ASO (*Soifer et al., 2012*) targeting both mouse and Human isoforms. Endotracheal administration of this ASO from day 7 in the bleomycin-induced lung fibrosis model allowed us to inhibit locally and efficiently *Prrx1* TFs during the active fibrosis phase in a 'curative' protocol. LNA-modified ASOs are protected from nuclease-mediated degradation, which significantly improves their stability and prolongs their activity in vivo. In comparison to earlier generations of ASO modifications, they have a massively increased affinity to their target RNA and their in vitro and in vivo activity does not depend on delivery reagents (*Soifer et al., 2012*) with potential translational applications in patients. As a proof of concept, we confirmed that intratracheal administration of *Prrx1*-specific ASO inhibited the upregulation of mouse PRRX1a and 1b

expression at both mRNA and protein level at day 14 in bleomycin-treated mice. Pulmonary fibrosis development was also reduced in these animals. While our in vitro findings in adult Human lung fibroblasts showed that PRRX1 inhibition mainly impacted ACTA2 expression levels, *Prrx1* ASO treatment in the bleomycin mouse model of lung fibrosis also inhibited the deposition of ECM proteins such as Collagen 1, Fibronectin, Tenascin C, and Elastin. This difference regarding ECM compound at day 14 may reflect the effect of *Prrx1* ASO on the overall fibrosis development; upon the proliferation / accumulation as well as impaired myofibroblastic differentiation of mesenchymal cells in vivo from the beginning of the ASO treatment at day 7. The impact of *Prrx1* ASO was found to be less pronounced in mice treated during the late fibrosis phase (between day 21 and 28). These findings suggest that *Prrx1* inhibition may be more effective during the active phase of fibrosis. However, inhibiting *Prrx1* during the late phase did not appear to affect lung repair in vivo, which is also supported by the lung alveosphere assay with *Prrx1* deficient MEFs. In addition, we confirmed the anti-fibrotic effect of the *Prrx1* ASO in a second model of fibrosis using ex vivo culture of Human or mouse PCLS stimulated with a cocktail of fibrosis-associated cytokines (*Lehmann et al., 2018*). A promising anti-fibrotic effect was also observed in IPF PCLS treated with *PRRX1* ASO. To confirm and study further these results obtained with *Prrx1* ASO, it will be interesting to perform in vivo CRE-mediated conditional and inducible loss of *Prrx1* (*Feldmann et al., 2021*) in the different mesenchymal lung cell lineages expressing this transcription factor in the bleomycin model of lung fibrosis as discussed above. Similarly, lineage tracing experiments in mice should be undertaken to identify the source of those *Prrx1*-positive mesenchymal cells during lung fibrosis development.

Targeting of others transcription factors such as GLI (*Moshai et al., 2014*), FOXM1 (*Penke et al., 2018*), FOXF1 (*Black et al., 2018*), FOXO3 (*Al-Tamari et al., 2018*), RUNX2 (*Mümmler et al., 2018*), and TBX4 (*Xie et al., 2016*) was also shown to inhibit fibrosis development in this mouse experimental model of pulmonary fibrosis. However, at the exception of TBX4 and PRRX1, the expression of all these other TFs is not restricted to mesenchymal lineages, which means that targeting those TFs may impact both lung fibrosis and epithelial regeneration/repair. Finally, PRRX1 inhibition as a potential therapeutic approach in fibrosis is not restricted to the lung. Adenoviral shRNA mediated inhibition of *Prrx1* in the thioacetamide model of liver fibrosis in rats also decreased fibrotic lesions, collagen deposition and hepatic stellate cells myofibroblastic differentiation (*Gong et al., 2017*).

In conclusion, our study unveils the role of the pro-fibrotic and mesenchyme associated PRRX1 TFs in lung fibrosis, involved in fibroblasts proliferation and TGF-β pathway responsiveness during myofibroblastic differentiation. Direct inhibition of PRRX1 transcriptional activity in mesenchymal cells may be a potential therapeutic target in IPF as supported by our encouraging results in Human IPF PCLS. Furthermore, the effectiveness of the administration of *Prrx1* ASO in the bleomycin model of pulmonary fibrosis is particularly interesting. The route of administration we used constitutes a first attempt to locally inhibit a pro-fibrotic TF. The possibility of a local administration of an antifibrotic is seductive: current antifibrotics, administered systemically, are burdened with significant adverse events, which significantly attenuates their effect on health-related quality of life (*Graney and Lee, 2018*). Inhaled pirfenidone and other inhaled compounds (*Kaminskas et al., 2019*) are currently investigated in IPF, but none are directly acting on a mesenchymal transcription factor. Although already proved effective in asthma (*Krug et al., 2015*), local transcription factor inhibition has never been investigated in IPF so far.

## Materials and methods

### Human lung samples

IPF lung samples were obtained from patients undergoing open lung biopsy or at the time of lung transplantation (n=39; median age 61 yr; range 51–70 yr). IPF was diagnosed according to 2011 ATS/ERS/JRS/ALAT criteria, including histopathological features of usual interstitial pneumonia (*Raghu et al., 2011*). Lung samples obtained after cancer surgery, away from the tumor, were used as controls; normalcy of control lungs was verified histologically (n=35 patients; median age 64 yr, range 28–83 yr).

### In vivo experiments

All experiments were performed using adult male C57BL/6 mice and intratracheal administration of saline or bleomycin, as previously described (*Moshai et al., 2014*). To investigate the involvement of

PRRX1 in fibrogenesis, mice were treated with third generation locked nucleic acid (LNA)-modified ASO targeting PRRX1 designed by Secarna Pharmaceuticals GmbH& Co, Planegg/Martinsried, Germany. The following sequence was used (+indicates an LNA modification, while * indicates a phosphorothioate (PTO) linkage) to target *Prrx1* (*Prrx1* ASO):+T*C*+A*+G*G*T*T*G*G*C*A*A*T*G*+C* +T*+G.

A previously published and validated (*Jaschinski et al., 2015*) negative control ASO (Cont ASO) was used:+C*+G*+T*T*T*A*G*G*C*T*A*T*G*T*A*+C*+T*+T.

Briefly, all mice were placed under isoflurane anesthesia during the endotracheal instillation and received 25 µL injections of either 20nmoles of ASO or PBS every other day. The knockdown efficiency of *Prrx1* ASO and lack of effects of control ASO were first validated in naive mice, which were treated with either *Prrx1* ASO or control ASO, compared to PBS-treated mice (data not shown). In the next set of experiments, the bleomycin group was instilled with control or *Prrx1* ASO while the saline group received PBS from day seven after the bleomycin injection. In the last set of experiments, the saline group was given control ASO or PBS from day 21, while the bleomycin group was instilled with control or *Prrx1* ASO. The lungs were harvested on day 14 or day 28, for further analysis (see *Figure 6A* and *Figure 7—figure supplement 1*). Hematoxylin, eosin and picrosirius staining were performed routinely to evaluate the morphology of the lung. Semiquantitative histological assessment of lung injury used the grading system described by Inoshima and colleagues (*Inoshima et al., 2004*). Total mRNA was extracted from mouse lung homogenates, and the expression of the genes of interest was quantified by real-time PCR, as previously described. Proteins were extracted from mouse lung homogenates and western blotting was performed by standard techniques as previously described (*Moshai et al., 2014*).

The *Prrx1* heterozygous mouse strain (129S-*Prrx1*^tm1Jfm/Mmmh (*Lu et al., 1999*), RRID:MMR-RC_000347-MU) was obtained from the Mutant Mouse Resource and Research Center (MMRRC) at University of Missouri (USA), an NIH-funded strain repository, and was donated to the MMRRC by Pr James Martin (Texas Agricultural and Mechanical University: Health Science Center, USA).

## Analysis of publicly available control and IPF lung RNA microarray and single-cell datasets

Analysis of curated NCBI GEO dataSet GDS1252 (GSE2052 *Wang et al., 2006*), GDS4279 (GSE24206 *Meltzer et al., 2011*) and GDS3951 (GSE21411 *Cho et al., 2011*) from control and IPF whole lung transcriptomic data was performed using NCBI "Compare 2 sets of samples" Geo Dataset analysis tool (https://www.ncbi.nlm.nih.gov/geo/info/datasets.html). The list of upregulated genes in IPF samples compared to control ones in the three transcriptomic datasets was established by generating a Venn's diagram (Venny 2.1 tool; http://bioinfogp.cnb.csic.es/tools/venny/). Gene annotation analysis (Protein Class) was then performed with PANTHER (*Mi et al., 2017*). Single-cell transcriptomic analysis of lungs from transplant donors or recipients with pulmonary fibrosis was performed using the dataset available at http://www.ipfcellatlas.com/ (*Adams et al., 2020*; *Habermann et al., 2020*). The analysis of control and bleomycin-treated mouse lungs (day 11, 14 and 21) was performed using the BioTuring software (Bioturing Inc, San Diego, USA) with the lung scRNAseq dataset available at https://www.fibroxplorer.com (*Buechler et al., 2021*).

## Gene expression profiling

Integrity of total RNAs was assessed using an Agilent BioAnalyser 2100 (Agilent Technologies) (RIN >9). A total of 150–200 ng of RNA samples (with RIN >9) were labeled with Cy3 dye using the low RNA input QuickAmp kit (ref 5190–2305, Agilent) as recommended by the supplier. After fragmentation step, 600 ng of Cy3-labeled cRNA probes were hybridized on 8x60K v3 high density SurePrint G3 gene expression human Agilent microarrays (G4851C, Agilent) with respect to the manufacter protocol. Statistical analysis and Biological Theme Analysis. Microarray data analyses were performed using the Bioconductor package limma. Briefly, data were normalized using the quantile method. No background subtraction was performed. Replicated probes were averaged after normalization and control probes removed. Statistical significance was assessed using the limma moderated t-statistic. All p-values were adjusted for multiple testing using the Benjamini-Hochberg procedure. Differentially expressed genes were selected based on an adjusted p-value below 0.05. Enrichment in biological themes (Molecular function, Upstream regulators and canonical pathways) were performed using

Ingenuity Pathway Analysis software (http://www.ingenuity.com/). Expression dataset supporting the *Figure 5C* and *Figure 5—figure supplement 3* of this study has been deposited in the Gene Expression Omnibus GSE161364: Impact of PRRX1 knockdown on the transcriptome of normal human lung fibroblasts (NHLF) in the presence or the absence of TGF-β (8 experimental conditions performed in duplicates).

## Cell culture experiments

Primary Human lung fibroblasts from control and IPF patients were derived from Human lung explants as previously described (*Brayer et al., 2017*). Primary HLFs were cultured with Dulbecco's modified Eagle's medium (Thermo-Fisher Scientific, Waltham, USA) supplemented with 10% fetal calf serum (FCS) and antibiotics and used at passage 4–6 as previously described (*Brayer et al., 2017*). Primary lung fibroblasts were treated for 48 hr with TGF-β1 (Peprotech, Neuilly sur seine, France), PGE2 (100 nM, Sigma, Saint-Louis, USA), NS398 (10 μM, Sigma Aldrich, Saint-Louis, USA) and Fasudil (35 μM, Sigma Aldrich, Saint-Louis, USA). Primary culture of alveolar lung macrophages and alveolar type 2 epithelial cells from control lung were performed as described previously (*Menou et al., 2018*). MEFs were prepared from E13.5 WT or *Prrx1$^{-/-}$* mouse embryos and cultured as described elsewhere (*Xu, 2005*).

### PRRX1 siRNA-mediated inhibition

primary HLFs were transfected at 50–60% confluency in 6- or 12-well plates. All transfection reactions were performed with, lipofectamine RNAimax (ThermoFisher Scientific, Waltham, USA) reagent transfection, in accordance with the manufacturer's instruction as described previously (*Brayer et al., 2017*). To suppress *PRRX1* endogenous expression, lung fibroblasts were transfected with two independent sequences targeting both *PRRX1a* and *PRRX1b* mRNA: siRNA#1, 5'-AGAAAGCAGCGA AGGAAUA-3' (GE Health Care-Dharmacon, Lafayette, USA) & siRNA#2, 5'-GCGAAGGAAUAGGACA ACCUUCAAU-3' (Thermo Fisher Scientific, Waltham, USA). siRNA Negative Control Low GC (12935–200, Invitrogen, Carlsbad, USA) was used as a negative control (siControl). A final concentration of 30 nM for each siRNA was used. A *PPM1A* siRNA smartpool was used to knock down *PPM1A* in lung fibroblasts (D-009574, GE Health Care-Dharmacon, Lafayette, USA).

### Fibroblast myofibroblastic differentiation

Serum-starved fibroblasts for 16 hr were cultured for 48 hr with *PRRX1* siRNA sequences or negative control, then treated during 48 hr with TGF-β1 (1 ng/ml). Cells were harvested and assayed for further mRNA and western Blot analysis (as described below).

Fibroblast proliferation was assayed 72 hr after *PRRX1* siRNA transfection transfection. The proliferation rate was measured by CyQUANT NF Cell Proliferation Assays (Thermofisher, Waltham, USA) which quantify cellular DNA content using a microplate reader (Infinite 200 PROTECAN, Männedorf, Suisse). Cytotoxicity was assayed by trypan blue exclusion assay.

### Stiffness experiments

Cells were plated during 48 hr on substrate with defined elastic modulo (EM) which correspond to the matrix rigidity: 1,5 kPa (soft matrix, 81291), 28 kPa (stiff matrix, 81191) or in the GPa range (glass, 81158) (ibidi, GmbH, Planegg, Germany). Coating with Fibronectin (Human plasma, Sigma Aldrich) was performed according to plate's manufacturer instructions (ibidi, GmbH, Planegg, Germany).

### Fibroblast-derived matrix experiments

Thick matrices were produced from primary Human control and IPF lung fibroblast using an adapted protocol from *Castelló-Cros and Cukierman, 2009*. Control primary lung fibroblasts were next seeded on these matrices and cultured for 48 hr as described in *Castelló-Cros and Cukierman, 2009*. Control primary lung fibroblasts were next seeded on this 3D matrices and treated with Imatinib (10 μg/ml), Nintedanib (10 nM) or PDGFR V inhibitor (10 nM) (all from Selleckchem, Boston USA) in DMEM medium supplemented with 2% of FCS. Forty-eight hr after treatment, cells were harvested for mRNA analysis (as described below).

*Lung alveosphere culture* in matrigel was performed following an adapted protocol from *Konishi et al., 2022* using wild-type and *Prrx1⁻ᐟ⁻* MEFs as stromal cells with wild-type primary mouse lung derived epithelial cells. Spheres were counted at day 15.

## Atomic Force Microscopy Measurements

AFM indentations were performed using an atomic force microscope (Nanowizard4, JPK Instruments-Bruker, USA), mounted on a Zeiss microscope (Axio Observer Z1, Zeiss, Germany) All measurements were performed at room temperature using 35 mm Petri dish (ibidi, Germany). Measures were performed using B1000-Cont cantilever (Nanoandmore, France). Cantilevers were calibrated using the thermal oscillation method. Samples were intended with a maximum force of 1 nN. The Hertz model was used to determine the elastic properties of the tissue. The upper 1 μm of tissue were considered for all fits. Tissue samples were assumed to be incompressible and a Poisson's ratio of 0.5 was used in the calculation of the Young's elastic modulus.

## mRNA analysis

Total mRNA was extracted from Human lung samples and from cultured fibroblasts, and cDNA was obtained by standard techniques as described (*Brayer et al., 2017*). *Ubiquitin C* (*UBC*) mRNA was used for normalization. Primers are reported in *supplementary file 3*.

## Western blot analysis

Proteins were extracted from Human lung samples and from cultured fibroblast by standard techniques as described (*Brayer et al., 2017*). Western blotting was performed under denaturing and reducing conditions. Antibodies are reported in *supplementary file 4*.

*Immunofluorescence*, cells were cultured in serum-free medium on Permanox Lab-Tek Chamber Slides (Nunc, ThermoFisher Scientific, Waltham, USA) as described (*Joannes et al., 2016*). A primary antibody against PRRX1 (see *supplementary file 4*) with an anti-Rabbit 568 Alexa-Fluor coupled secondary antibodies were used after PFA fixation as described (*Joannes et al., 2016*). Alexa-fluor 488 coupled Phalloidin was used to stain actin fibers and DNA was stained with DAPI (Invitrogen, Carlsbad, USA).

*Chromatin immunoprecipitation*, ChIP analyses were performed with Abcam's ChIP Kit –' One Step' (ab117138) following manufacturer instructions. Briefly, formaldehyde cross-linked chromatin from NHLF (primary control lung Human fibroblasts purchased from Lonza, Basel, Switzerland) was prepared as previously described (*Brayer et al., 2017*). Chromatin fragments (from about 1x10⁶ cells) were immunoprecipitated with antibodies (see *supplementary file 4*) against PRRX1, RNA POL-II and a control IgG (both from Abcam's 'ChIP Kit - One Step'). The precipitated DNA was amplified by real-time PCR with primer sets designed to amplify PRRX1 response elements (PRE) in *CCNA2*, *CCNE2*, *MKI67* and *PPM1A* loci and *GAPDH* TSS (*supplementary file 1*). The results are expressed relative to RNA POL-II occupancy of a given locus. Putative PRE sequences in those different loci were first identified within the regulatory build / core promoter sequences (NHLF cells) available from Ensembl database (*Zerbino et al., 2018*) using FIMO (MEME Suite Programs *Bailey et al., 2009*) and LASAGNA-Search (*Lee and Huang, 2013*) tools with PRE consensus sequence (Jaspar database *Wasserman and Sandelin, 2004*).

## Culture of ouse and human precision-cut lung slices (PCLS) with fibrosing cocktail (FC)

Mouse and Human PCLS were obtained from 6 mm biopsy punch performed on agarose-filled lung samples. Mouse PLCS were obtained from 8-week-old C57BL/6 mice. Control and IPF Human PCLS were prepared respectively from lung samples obtained after cancer surgery, away from the tumor or from transplant recipients with pulmonary fibrosis. Lung cylinders were next sliced (300 μm) using a McIlwain Tissue Chopper and processed as previously described (*Alsafadi et al., 2017*; *Lehmann et al., 2018*). Briefly, control Human PCLS were treated with FC for 48 hr in presence of *PRRX1* ASO (10 μM) or control ASO (10 μM) without transfection reagent. Mouse PCLS were first stimulated with FC for 48 hr then treated with either control or *PRRX1* ASO for additional 48 hr without transfection reagent. FC was prepared as previously described (*Alsafadi et al., 2017*) and all the cytokines were

purchased from PeproTech (Neuilly sur Seine, France). In basal condition, mouse PCLS, control or IPF Human PCLS were treated only with control ASO (10 µM) for 48 h. PCLS were harvested for mRNA, protein and histological studies using previously described methods (*Moshai et al., 2014*). Picrosirius staining and quantification were performed as described previously (*Justet et al., 2017*).

## Flow cytometry

FACS analysis was assayed after 72 hr siRNA PRRX1 transfection. Cells were harvested and prepared as previously described (*Kim and Sederstrom, 2015*). Cells were stained for propidium iodide (0.5 mg/ml) (Sigma, Saint-Louis, USA) and anti Ki-67 Alexa Fluor 488 (1/500) or isotype control IgG1 Alexa Fluor 488 (1/500) (Becton, Dickinson and Company (BD), Franklin Lakes, US). Samples were analyzed by flow cytometry with a FACSCanto II flow cytometer using the FACSDIVA software (Becton Dickinson, New Jersey, USA) for data acquisition (15,000–30,000 events) and the FlowJo, 9.9.6 software (FlowJo LLC, Ashland, Oregon, USA) for analysis.

## Immunohistochemistry and hydroxyproline assay

Immunohistochemistry, paraffin-embedded sections (n=5 per group) were treated as described (*Joannes et al., 2016*). Antibodies are reported in *supplementary file S4*. To validate the specificity of immunostaining, antibodies were replaced by a matched control Isotype. All digital images of light microscopy were acquired with a DM400B microscope (Leica) equipped with a Leica DFC420 CDD camera and analyzed with Calopix software (TRIBVN). Measurement of PRRX1, ACTA2 or PDGFR-positive areas relative to total lung section surface (regarding ACTA2 and PDGFR IHC) or hematoxylin area (as total nuclear signal quantification for PRRX1 IHC) was performed as described previously (*Justet et al., 2017*). With respect to histological quantifications, areas of interest were chosen based on the presence of major alveolar thickening as well as fibrous changes and masses (confirmed by picrosirius staining on serial-sections).

Double immunohistochemistry for PRRX1 with Vimentin or ACTA2 were performed as previously described (*Chen et al., 2010*) using the Citrate 'Heat Induced Epitope Retrieving' (HIER) microwave method between the consecutive stainings. Briefly, ACTA2 or Vimentine stainings were first revealed with 'Dako Permanent Red Substrate-Chromogen' (Agilent) after using respectively Rabbit or Mouse 'Histofine Simple Stain AP' (Nicheiri Biosciences) as secondary antibodies. PRRX1 staining was next revealed using 'Dako Liquid DAB +Substrate Chromogen System' (Agilent) after using Rabbit 'Histofine Simple Stain MAX PO' as secondary antibody (Nicheiri Biosciences). With respect to KI67 and PDGFR double stainings on mouse lung tissues, KI67 staining was first revealed with 'Dako Liquid DAB +Substrate Chromogen System' (Agilent) after using a Rabbit 'Histofine Simple Stain MAX PO' (Nicheiri Biosciences) as secondary antibody. PDGFR staining was next revealed using 'Vector Blue Substrate alkaline phosphatase' (Vector Lab) after using Rabbit 'Histofine Simple Stain AP' as secondary antibody (Nicheiri Biosciences). Nuclei were counterstained with Nuclear Fast Red (Vector labs) and the slides mounted with Vectamount Permanent mounting medium (Vector labs).

Hydroxyproline assay, total collagen and protein quantifications were performed using respectively the Quickzyme Biosciences (Leiden, The Netherlands) hydroxyproline assay and total protein assay kits from paraffine lung sections (15 sections of 10 µm per samples), following manufacturer instructions. Hydroxyproline assay was also performed on whole lung extract using the inferior lobe as described previously (*Justet et al., 2022*).

## Statistical Analysis

Most data are represented as dot plots with median, unless specified. All statistical analysis were performed using Prism 8 (GraphPad Software, La Jolla, CA). No data were excluded from the studies and for all experiments, all attempts at replication were successful. For each experiment, sample size reflects the number of independent biological replicates and is provided in the figure legend. We used non-parametric Mann-Whitney U test for comparison between two experimental conditions. Paired data were compared with Wilcoxon signed-rank test. We used non-parametric Kruskall Wallis test followed by Dunn's comparison test for group analysis. Comparison of histological scores on Day 14 was performed with Fisher's exact test. A $P$-value <0.05 was considered to be statistically significant. Exact $P$ values and definition and number of replicates are given in the respective figure legend.

## Study approval

The study on human material was performed in accordance with the Declaration of Helsinki and approved by the local ethics committee (CPP Ile de France 1, No.0811760). Written informed consent was obtained from all subjects. All animal experiments were conducted in accordance with the Directive 2010/63/EU of the European Parliament and approved by the local Animal ethics committee ("Comité d'éthique Paris Nord n°121", APAFiS #4778 Etudedufacteurdetran_2016031617411315).

## Acknowledgements

We thank Olivier Thibaudeau and Laure Wingertsmann (Morphology Platform, Inserm U1152 of X Bichat Medical School, Paris) for their efficient collaboration and the Flow Cytometry Platform of Inserm U1149 (CRI, X Bichat Medical School, Paris) as well. We also thank Jan Willem Duitman (Academic Medical Center, Amsterdam, The Netherlands), Alberto Baeri (IPMC, France), Nicolas Nottet (C3M, Nice, France), Kevin Lebrigand (UCA genomics platform, IPMC, France), Laurent Boyer and Gregoire Justeau (IMRB Créteil, France) for their technical help.

## Additional information

### Competing interests

Antoine Froidure: received an unrestricted research grant from Boehringer Ingelheim, consulting fees from Boehringer Ingelheim and payment or honoraria from Boehringer Ingelheim and Roche that were paid to their institution. The author has no other competing interests to declare. Ksenija Schirduan: was a former employee of Secarna Pharmaceuticals. Frank Jaschinski: is an employee of Secarna Pharmaceuticals. Bruno Crestani: received grants from Boehringer Ingelheim, received consulting fees, payment or honoraria, and/or support for meetings and/or travel from Apellis, Astra Zeneca, BMS, Boehringer Ingelheim, Novartis, Sanofi. BC also participates on a data safety monitoring board or advisory board for Apellis, BMS, Boehringer Ingelheim, Sanofi and is a member on the Member of the board of trustee of the Fondation du Souffle. The author has no other competing interests to declare. The other authors declare that no competing interests exist.

### Funding

| Funder | Grant reference number | Author |
|---|---|---|
| Agence Nationale de la Recherche | JCJC ANR-16-CE14-00 | Arnaud A Mailleux |
| European Respiratory Society | ERS-LTRF 2015 - 4476 | Antoine Froidure |
| Fondation pour la Recherche Médicale | FDT2021060129750 | Méline Homps-Legrand |
| Fondation pour la Recherche Médicale | FDM41320 | Aurélien Justet |

The funders had no role in study design, data collection and interpretation, or the decision to submit the work for publication.

### Author contributions

Emmeline Marchal-Duval, Madeleine Jaillet, Mada Ghanem, Andreas Gunther, Bruno Crestani, Arnaud A Mailleux, Conceptualization, Resources, Data curation, Formal analysis, Supervision, Funding acquisition, Validation, Investigation, Visualization, Methodology, Writing – original draft, Project administration, Writing – review and editing; Méline Homps-Legrand, Conceptualization, Resources, Data curation, Formal analysis, Supervision, Funding acquisition, Validation, Investigation, Methodology, Writing – original draft, Project administration, Writing – review and editing; Antoine Froidure, Conceptualization, Investigation, Methodology, Writing – original draft, Writing – review and editing; Deneuville Lou, Conceptualization, Data curation, Formal analysis, Supervision, Funding acquisition, Validation, Investigation, Visualization, Methodology, Writing – original draft, Project administration,

Writing – review and editing; Aurélien Justet, Arnaud Maurac, Emilie Fortas, Investigation, Methodology, Writing – review and editing; Aurelie Vadel, Aurelie Cazes, Audrey Joannes, Laura Giersh, Investigation; Herve Mal, Pierre Mordant, Marin Truchin, Carine M Mounier, Resources, Investigation; Tristan Piolot, Ksenija Schirduan, Martina Korfei, Resources, Investigation, Methodology; Bernard Mari, Resources, Investigation, Methodology, Writing – original draft, Writing – review and editing; Frank Jaschinski, Conceptualization, Resources, Investigation, Methodology

### Author ORCIDs
Andreas Gunther ⓘ http://orcid.org/0000-0002-2187-0975
Bernard Mari ⓘ http://orcid.org/0000-0002-0422-9182
Arnaud A Mailleux ⓘ http://orcid.org/0000-0003-4191-1778

### Ethics
Human subjects: The study on human material was performed in accordance with the Declaration of Helsinki and approved by the local ethics committee (CPP Ile de France 1, No.0811760). Written informed consent was obtained from all subjects.

All animal experiments were conducted in accordance with the Directive 2010/63/EU of the European Parliament and approved by the local Animal ethics committee ("Comité d'éthique Paris Nord 121", APAFiS #4778 Etudedufacteurdetran_2016031617411315).

### Decision letter and Author response
Decision letter https://doi.org/10.7554/eLife.79840.sa1
Author response https://doi.org/10.7554/eLife.79840.sa2

---

## Additional files

### Supplementary files
• Supplementary file 1. List of common up-regulated genes in all three IPF transcriptome database analyzed.

• Supplementary file 2. Attenuation of the TGF-β response following PRRX1 knock-down. The table contains the best 597 genes significantly modulated by TGF-β in NHLF treated with siControl#1 and #2 (log2 fold change >4; adj. P Val <0.05). The table indicates the Fold Change (FC) for each of the genes: (i) column 1: TGF-β1" stimulation in presence of siControl#1; column 2: TGFβ1 fold stimulation in presence of siRNA#1; column 4: TGFβ1 fold stimulation in presence of siControl#2; column 5: TGFβ1 fold stimulation in presence of siRNA#2". Columns 5 and 6 give the percentage of residual modulation by TGF-β for each PRRX1 siRNA. Modulations are shown in progressively brighter shades of blue (attenuation) and orange (over-activation). The mean residual fold change following PRRX1 KD is 56.6% (siRNA#1) and 80.0% (siRNA#2).

• Supplementary file 3. PCR primer sequences.

• Supplementary file 4. Antibody list.

• MDAR checklist

### Data availability
For gene expression profiling, publicly available datasets were obtained from NCBI Gene Expression Omnibus (GSE2052, GSE24206 and GSE21411) , IPF Cell Atlas (http://www.ipfcellatlas.com) or FibroXplorer (https://www.fibroXplorer.com). Newly generated expression dataset has been deposited in the Gene Expression Omnibus GSE161364. All data generated or analyzed during this study are included in the manuscript and supporting files.

The following dataset was generated:

| Author(s) | Year | Dataset title | Dataset URL | Database and Identifier |
|---|---|---|---|---|
| Mari B, Mounier C | 2023 | Impact of PRRX1 knockdown on the transcriptome of normal human lung fibroblasts (NHLF) in the presence or the absence of TGF-β | https://www.ncbi.nlm.nih.gov/geo/query/acc.cgi?acc=GSE161364 | NCBI Gene Expression Omnibus, GSE161364 |

The following previously published datasets were used:

| Author(s) | Year | Dataset title | Dataset URL | Database and Identifier |
|---|---|---|---|---|
| Pardo A, Gibson KF, Selman M, Kaminski N | 2005 | IPF versus Control | https://www.ncbi.nlm.nih.gov/geo/query/acc.cgi?acc=GSE2052 | NCBI Gene Expression Omnibus, GSE2052 |
| Meltzer E | 2011 | Validated Gene Expression Signatures of Idiopathic Pulmonary Fibrosis | https://www.ncbi.nlm.nih.gov/geo/query/acc.cgi?acc=GSE24206 | NCBI Gene Expression Omnibus, GSE24206 |
| Cho JH, Gelinas R, Wang K, Etheridge A | 2011 | Systems biology of interstitial lung diseases | https://www.ncbi.nlm.nih.gov/geo/query/acc.cgi?acc=GSE21411 | NCBI Gene Expression Omnibus, GSE21411 |

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
