## [Editor Report]

This manuscript is of interest to scientists in the field of tissue injury and repair. It provides novel molecular mechanisms of a transcription factor, Prrx1, in fibroblast activation following lung injury. Overall, the work suggests that PRRX1 plays a functional role downstream of TGFb1 to elicit some aspects of the fibrotic response and that PRRX1 could represent an important therapeutic target to treat fibrosis.

---

## [Decision Letter]

**Decision letter after peer review:**

Thank you for submitting your article "Identification of Paired-related Homeobox Protein 1 as a key mesenchymal transcription factor in Pulmonary Fibrosis" for consideration by *eLife*. Your article has been reviewed by 2 peer reviewers, and the evaluation has been overseen by a Reviewing Editor and Paul Noble as the Senior Editor. The reviewers have opted to remain anonymous.

Essential revisions:

The strengths of this work are the multiple approaches applying human and mouse lung tissue used by the authors to test the role of PRRX1 in lung fibrosis, however, major limitations reduce the quality and potential impact of the manuscript in its current stage. Both reviewers raise important concerns that should be fully addressed. Specifically, please note the following:

1) The discrepancy between the strong in vivo phenotype and the more in vitro data needs to be addressed, such as investigating other potential mechanisms.

2) Cell type specificity needs to be investigated, especially with respect to changes in proliferation and differentiation.

3) The finding that Prrx1 is also found in normal lung fibroblasts and downregulated by TGFb needs to be better explained in putting into the contact of fibrosis.

4) Proper control groups need to be included.

Add additional experimental data and control groups, to fully explain the phenotypes observed. In addition, the manuscript text needs to be tempered down with respect to the conclusion that can be drawn.

*Reviewer #1 (Recommendations for the authors):*

The authors used siRNA to knockdown Prrx1 for primary fibroblasts, while they used ASO in in vivo experiments. Does Prrx1 ASO more efficiently knockdown Prrx1 if used in in vitro?

The authors mentioned that Prrx1 is expressed in Col14a1+ fibroblasts in normal lungs according to Xie et al. 2018. More recent studies showed that Col14a1+ fibroblasts in the lung are Pi16+ adventitial fibroblasts that reside in bronchovascular bundles, which are distinct from fibroblasts that arise in injured lungs. Addressing if Prrx1 is upregulated in injured-lung specific fibroblasts by using other data sets (Tsukui et al. 2020 Nature Communications, or Redente et al. 2020 JCI Insight) may help understand the in vivo regulation of Prrx1.

Figure S9A Immunofluorescence. The image background signal is different between WT and KO. In the WT there is background fluorescence in the Prrx1 channel, while in the KO there is almost no background signal. Please process the two samples in the same way.

*Reviewer #2 (Recommendations for the authors):*

Figure 1: Identification of PRRX1 as a transcription factor reactivated in IPF lung.

It would be interesting to have more clinical data on the group of IPF lungs expressing high levels of PRRX1 (represented by the 5 dots in Figure 1C) compared to the IPF lungs expressing a lower level of PRRX1. Are the high expressors representative of more severe clinical outcomes? Quantification of the PRRX1-positive cells In IPF and Donor lungs in pictures shown in D would also be great.

Figure 2: PRRX1 is a mesenchymal transcription factor upregulated in primary Human lung IPF fibroblasts.

A violin plot for the expression of PRRX1 in Donor and IPF fibroblasts shown in A would be great. The expression of PRRX1 seems to be also in Donor fibroblasts (open circles) and not only in lung fibrosis (closed circles). This does not fit necessarily with the data shown in C. The authors should discuss this difference and also propose a function for PRRX1 in donor fibroblasts.

Figure 3: PRRX1 is modulated by growth factors and matrix environment.

Figure 3A: the authors need to look at different time points to find out when the negative feedback loop leading to the decrease in PRRX1 expression following TGFb1 treatment starts. Figure 3B: It was expected that the stiffer ECM (28 kPA\a) should have triggered a higher level of PRXX1 expression compared to the soft ECM (1.5 kPa). However, the authors observe the opposite and should offer some interpretation. Figure 3D and E need to be better explained especially the quantification for Figure 3D which seems to suggest that there is no difference in the elastic modulus of the IPF ECM vs Control ECM. Was this result expected? Figure 3E: the interpretation and the significance of the data shown are not clear.

Figure 4: PRRX1 knockdown decreased cell proliferation.

A supplementary panel showing Edu staining would have been great to confirm the decrease in proliferation triggered by the silencing of PRRX1 expression. Also IF showing the expression of PRRX1 by IF would be additional evidence for the loss of PRRX1 expression

Figure 5: PRRX1 inhibition decreased myofibroblast differentiation upon TGF-β1stimulation. This is a very nice piece of data.

Figure 6: PRRX1 inhibition attenuates lung fibrosis in the bleomycin murine model.

A schematic showing the experimental approach should be provided (treatment with ASO stating at d7 post bleo and analysis at d14). Another time window (ASO at d14 post bleo and analysis at day 28 should be provided). If possible, it would be also interesting to validate these data by measuring lung function in ASO control and Prrx1 animals.

Figure 7: PRRX1 inhibition decreases fibrosis markers in bleomycin mice.

As Prrx1 is also expressed in the adult lung mesenchyme, it would be interesting to determine what is the impact of silencing Prrx1 expression during homeostasis. Would the decrease in Prxx1 expression lead to a change in the activity of the resident mesenchymal niche (Cd45-Cd31-EpCam-Sca1+ cells)? This can be easily tested using alveolospheres (co-culture of normal mature AT2s with either rMC coming from ASO control or ASO Prrx1 lungs). If the mesenchyme from ASO Prrx1 lung is more efficient in promoting alveolosphere formation and growth, this would argue that the cells are less prone to adopt a myofibroblast-like phenotype when Prrx1 is silenced.

Supplemental Figure S2: PRRX1 expression profiles at single-cell resolution using the "IPF Cell Atlas" web database (http://ipfcellatlas.com/).

Figure S2C: Increase in PRRX1 expression is observed in Myofibroblasts in IPF vs. Control as expected.

However, there is also a significant increase in PRRX1 expression in the PLIN2+ fibroblasts in IPF vs. Control. What is the significance of this increase? Could it be that this leads to the loss of mesenchymal niche activity for AT2 cells (see comments for figure 7)?

Supplemental Figure S10: PRRX1 is increased during the fibrotic phase in mice with bleomycin-induced fibrosis.

What is happening to PRRX1 expression from d14 to d28 (during fibrosis resolution… Is PRRX1 expression decreasing)?

Supplemental Figure S15: a summary sketch of PRRX1 regulation and functions in lung fibroblasts.

The model is not clear as we would expect PRRX1 to first be upregulated in the context of TGFb1 signaling and then downregulated at a later stage due to the establishment of a negative feedback loop. In addition, PRRX1 should be located downstream of TGFBR2?

---

## [Author Response]

Essential revisions:The strengths of this work are the multiple approaches applying human and mouse lung tissue used by the authors to test the role of PRRX1 in lung fibrosis, however, major limitations reduce the quality and potential impact of the manuscript in its current stage. Both reviewers raise important concerns that should be fully addressed. Specifically, please note the following:1) The discrepancy between the strong in vivo phenotype and the more in vitro data needs to be addressed, such as investigating other potential mechanisms.

In the revised manuscript, we now investigate additional mechanisms in order to reconciliate the in vitro and in vivo data. The take home message would be that PRRX1 TFs influence cell proliferation in addition to TGF-β1 responsiveness and that inhibiting PRRX1 would impact both processes in vivo and then fibrogenesis.

2) Cell type specificity needs to be investigated, especially with respect to changes in proliferation and differentiation.

We performed additional experiments to better investigate the changes in proliferation in *Prrx1* ASO treated bleomycin mice showing a decrease in fibroblast proliferation in those animals.

3) The finding that Prrx1 is also found in normal lung fibroblasts and downregulated by TGFb needs to be better explained in putting into the contact of fibrosis.

In the revised manuscript, we performed additional experiments to better understand (timewise) the downregulation of *PRRX1* in response to TGF-β1 in lung fibroblast.

4) Proper control groups need to be included.Add additional experimental data and control groups, to fully explain the phenotypes observed.

In the revised manuscript, we now mention the experiments that we performed in naive mice including control groups. With respect to the in vivo experiments initially presented, we had to restrict the overall number of animals to comply to institutional rules regarding animal uses. Nevertheless, we were authorized to include additional control groups in the new in vivo experiments targeting *Prrx1* during the late fibrosis phase between day 21 and 28.

In addition, the manuscript text needs to be tempered down with respect to the conclusion that can be drawn.

We modified the text accordingly to temper down our conclusions as suggested.

Reviewer #1 (Recommendations for the authors):The authors used siRNA to knockdown Prrx1 for primary fibroblasts, while they used ASO in in vivo experiments. Does Prrx1 ASO more efficiently knockdown Prrx1 if used in in vitro?

As suggested by the reviewer, we assayed in vitro the efficiency of the *Prrx1* ASO in primary lung fibroblasts (see Author response image 1). The *Prrx1* ASO did not seem to more efficiently knockdown PRRX1 *TFs* compared to the two *PRRX1* siRNA in our in vitro studies (as compared to Figure 4A-B).

**Author response image 1. sa2fig1:** K. nock down efficiency of Prrx1 ASO in primary control and IPF lung fibroblasts. (A) Dot plots with median showing PRRX1a (black) and PRRX1b (white) mRNA expression relative to the Control ASO condition (red dashed line) in control (circle) or IPF (square) primary lung fibroblasts (n=4 per group) treated for 72h with Prrx1 ASO (B) Immunoblot showing PRRX1 expression (n=4) in control and IPF fibroblasts treated 72h with control or Prrx1 ASO. The quantification of PRRX1 expression relative to GAPDH (loading control) is displayed as dot plot with median. Abbreviations: Control (Cont), Antisense oligonucleotide (ASO).

The authors mentioned that Prrx1 is expressed in Col14a1+ fibroblasts in normal lungs according to Xie et al. 2018. More recent studies showed that Col14a1+ fibroblasts in the lung are Pi16+ adventitial fibroblasts that reside in bronchovascular bundles, which are distinct from fibroblasts that arise in injured lungs. Addressing if Prrx1 is upregulated in injured-lung specific fibroblasts by using other data sets (Tsukui et al. 2020 Nature Communications, or Redente et al. 2020 JCI Insight) may help understand the in vivo regulation of Prrx1.

To investigate whether *Prrx1* was upregulated in lung-specific fibroblasts following bleomycin injury, we utilized the FibroXplorer database (Buechler et al., Nature 2021), which aggregates the datasets from Tsukui et al. (2020) in Nature Communications and Xie et al. (2018) in Cell Reports. In control lungs, *Prrx1* was mainly detected in *Pi16*-positive adventitial and *Col15a1* parenchymal “universal” fibroblast populations which also expressed *Col14a1*. Notably, *Prrx1* was still detected in the fibrosis-associated *Lrrc15-*positive fibroblast population (which also showed positivity for the recently identified *Cthrc1* fibrosis marker by Tsukui et al. 2020 Nature Communications) in animals treated with bleomycin, as shown in Figure 6—figure supplement 1 (see also lines 393-403 in revised manuscript).

Figure S9A Immunofluorescence. The image background signal is different between WT and KO. In the WT there is background fluorescence in the Prrx1 channel, while in the KO there is almost no background signal. Please process the two samples in the same way.

We thank the reviewer for pointing this discrepancy, which is now corrected in the revised figure 5-supplement figure 5.

Reviewer #2 (Recommendations for the authors):Figure 1: Identification of PRRX1 as a transcription factor reactivated in IPF lung.It would be interesting to have more clinical data on the group of IPF lungs expressing high levels of PRRX1 (represented by the 5 dots in Figure 1C) compared to the IPF lungs expressing a lower level of PRRX1. Are the high expressors representative of more severe clinical outcomes?

Analysis of clinical data available (lung function and mortality) did not reveal any clear association between *PRRX1* expression levels and clinical outcomes (even in these five “high expressors”). Insufficient statistic power related to the actual number of these patients could explain.

Quantification of the PRRX1-positive cells In IPF and Donor lungs in pictures shown in D would also be great.

Quantification of the upregulation of PRRX1 positive cells in IPF lungs compared to control ones by immunohistochemistry is displayed in Figure 1-supplement figure 1. We revised the manuscript accordingly (see lines 136-138).

Figure 2: PRRX1 is a mesenchymal transcription factor upregulated in primary Human lung IPF fibroblasts.A violin plot for the expression of PRRX1 in Donor and IPF fibroblasts shown in A would be great. The expression of PRRX1 seems to be also in Donor fibroblasts (open circles) and not only in lung fibrosis (closed circles). This does not fit necessarily with the data shown in C. The authors should discuss this difference and also propose a function for PRRX1 in donor fibroblasts.

We were unable to extract data on *PRRX1* expression from the Reyfman dataset using the "www.nupulmonary.org/resources/" website, so we substituted this panel with UMAP and violin plots that had previously been displayed in supplemental Figure S2. These plots were generated from the Banovich/Kropski datasets, which included both control and fibrotic lungs. In the Banovich/Kropski dataset, PRRX1 expression was predominantly associated with adventitial fibroblasts in control human lungs (see lines 154-155).

Accordingly, we agree with the reviewer that control fibroblasts express PRRX1, as supported by the lines of evidence presented in Figures 1 and 2. As stated in the manuscript regarding Figure 2C: "in vitro, both isoforms of *PRRX1* mRNA (*PRRX1a* and *PRRX1b*) were found to be strongly expressed by primary control lung fibroblasts compared to primary alveolar epithelial type 2 cells (AECII) and alveolar macrophages (Figure 2C) as determined by quantitative PCR (qPCR)" (lines 157-162). Moreover, we investigated the function of PRRX1 in control fibroblasts using siRNA, as shown in Figures 4 and 5. Our results demonstrated that PRRX1 TFs promote the proliferation of these primary lung fibroblasts and are also involved in their myofibroblastic differentiation in the presence of TGFb1.

Figure 3: PRRX1 is modulated by growth factors and matrix environment.Figure 3A: the authors need to look at different time points to find out when the negative feedback loop leading to the decrease in PRRX1 expression following TGFb1 treatment starts.

As suggested by the reviewer, we performed a time course analysis of PRRX1 isoform expression levels and showed that PRRX1 was downregulated only after 48h. This late downregulation of PRRX1 in response to TGF-b1, could be the signature of a negative feedback loop to limit cell-responsiveness to TGF-b1 when lung fibroblasts are fully differentiated into myofibroblasts at 48h as discussed in the revised manuscript (see Figure 9)

Figure 3B: It was expected that the stiffer ECM (28 kPA\a) should have triggered a higher level of PRXX1 expression compared to the soft ECM (1.5 kPa). However, the authors observe the opposite and should offer some interpretation.

Overall, our in vitro findings suggest that myofibroblastic differentiation in lung fibroblasts is associated with downregulation of *PRRX1* TFs. Interestingly, while the inhibitory effect of a stiff substrate (both basal plastic and 28kPa conditions) on PRRX1 levels is not complete, it is known that substrate stiffness regulates the degree of myofibroblastic differentiation. Therefore, the observed upregulation of PRRX1 on a soft substrate could reflect a potentiation effect associated with decreased myofibroblastic potential, whereas the decrease on a stiff substrate could indicate a potential feedback loop or rheostat, as has been observed with TGF-b1 (although in the latter case, the stimulus was stronger).

Figure 3D and E need to be better explained especially the quantification for Figure 3D which seems to suggest that there is no difference in the elastic modulus of the IPF ECM vs Control ECM. Was this result expected? Figure 3E: the interpretation and the significance of the data shown are not clear.

We thank the reviewer for highlighting that the purpose and significance of the experiments involving fibroblasts-derived ECM needed further clarification. In the revised manuscript, we now state that: “this 3D in vitro model will enable the study of how the interactions between lung fibroblasts and control or IPF stroma may affect PRRX1 expression levels.” for example (see lines 211-212). We further clarified this section of the paper by adding introductory sections and additional intermediary conclusions (as described in line 217-218 and lines 223-224).

As pointed out by the reviewer, we observed no significant difference in the elastic modulus between control and IPF-fibroblast derived ECM. Although we had no preconceived expectations about these results, we performed these measurements to determine whether the observed effects were driven by differences in ECM stiffness (as tested in Figure 3C-D) or were a consequence potential of variations in the ECM itself. With respect to interpretation and the significance of the data*,* “Our findings indicate that constitutive components of the microenvironment may play a role in controlling the expression of PRRX1 transcription factors in control fibroblasts (through PDGFR activation), regardless of local stiffness.” (lines 236-238) and “These results suggested that components present in IPF ECM directly modulated the expression of PRRX1 TFs in control fibroblasts independently of ECM stiffness. Further research will be necessary to determine the mechanisms that activate PGDFR by IPF-fibroblast derived ECM, leading to upregulation of PRRX1 TFs in control fibroblasts” (lines 551-555) as described in the revised manuscript.

In conclusion, the experiments featured in Figure 3 showed that PRRX1 TF mRNA levels were regulated by both ECM origin and stiffness in control fibroblasts, while only ECM stiffness modulated PRRX1 expression in IPF fibroblasts. Further investigation is also necessary to fully understand the mechanisms underlying the interactions between IPF fibroblasts and ECM origin, as well as a comprehensive study of the ECM components secreted by lung fibroblasts. These topics will be the focus of future studies.

Figure 4: PRRX1 knockdown decreased cell proliferation.A supplementary panel showing Edu staining would have been great to confirm the decrease in proliferation triggered by the silencing of PRRX1 expression. Also IF showing the expression of PRRX1 by IF would be additional evidence for the loss of PRRX1 expression

In the revised manuscript, we conducted a BrdU incorporation assay on control and IPF fibroblasts (see Figure 4—figure supplement 1). Our results showed that *PRRX1* siRNA treatment led to decreased BrdU incorporation in both control and IPF lung fibroblasts, compared to control siRNA treatment. All experiments presented in Figure 4 utilized quantitative methods, such as qPCR and immunoblot, to confirm the downregulation of PRRX1 (unfortunately, PRRX1 antibodies were not compatible with FACS analysis).

Figure 5: PRRX1 inhibition decreased myofibroblast differentiation upon TGF-β1stimulation. This is a very nice piece of data.Figure 6: PRRX1 inhibition attenuates lung fibrosis in the bleomycin murine model.A schematic showing the experimental approach should be provided (treatment with ASO stating at d7 post bleo and analysis at d14). Another time window (ASO at d14 post bleo and analysis at day 28 should be provided). If possible, it would be also interesting to validate these data by measuring lung function in ASO control and Prrx1 animals.

In the revised manuscript, we extended our investigation to assess the potential effect of *Prrx1* inhibition during the late fibrosis phase after bleomycin treatment at day 28, treating the animals with either control or *Prrx1* ASO every other day between day 21 and day 27. We selected this time window because the protocol approved by our local animal ethical committee only allowed for one week of ASO treatment, and we aimed to target the rebound in PRRX1 TF expression observed between day 21 and 28 (as discussed in response to comment #14). Interestingly, we found that the effects of *Prrx1* inhibition during the late fibrosis phase were less (but still) potent compared to *Prrx1* inhibition during the active/early fibrotic phase (see new Figure 7—figure supplement 1). Unfortunately, we were unable to validate the data by measuring lung function in control and Prrx1 ASO animals at this time. The necessary devices are not available in our animal facility. As suggested by the reviewer, schematics showing the experimental approaches were added in Figure 6 and Figure 7—figure supplement 1.

Figure 7: PRRX1 inhibition decreases fibrosis markers in bleomycin mice.As Prrx1 is also expressed in the adult lung mesenchyme, it would be interesting to determine what is the impact of silencing Prrx1 expression during homeostasis.

In the revised manuscript, we have updated our findings to include the following statement (lines 417-420): “After treating naive mice with PBS, control, or Prrx1 ASO every other day for a week, we observed no evidence of inflammation or upregulation of fibrosis markers associated with Prrx1 inhibition in healthy lungs, when compared to the PBS and control ASO groups (data not shown)”. As shown in Author response image 2, our data, suggest that transient *Prrx1* silencing did not have an impact on lung homeostasis with respect to fibrosis lesions and markers. In the future, it would be more relevant to investigate the effects of conditional genetic models to inhibit *Prrx1* in adult mouse lungs, since constitutive *Prxx1* loss of function is lethal at birth. Such investigations may provide a more comprehensive understanding of the role of Prrx1 in lung homeostasis.

**Author response image 2. sa2fig2:** ASO-mediated Prrx1 inhibition is not associated with an increase in fibrosis markers in healthy mouse lung. (A) Dot plots with median showing the mRNA expression of Prrx1a (black) and Prrx1b (white) isoforms at day 7 in wild-type mice treated every other day with PBS (circle), control ASO (square) or Prrx1 ASO (triangle); n=5 per group. Note the inhibition of Prrx1a and Prr1b mRNA expression levels in Prrx1 ASO treated mice compared to the PBS and control ASO groups. (B) Representative hematoxylin – eosin staining images (n=5 per group) showing no histological differences between wild type mice treated with either PBS, control or Prrx1 ASO at day 7. High magnifications are displayed on the right of the main panels. (C) Dot Plot with median showing the Collagen lung content (µg per lobe) at day 7 as measured by hydroxyproline in wild type mice treated with either PBS (circle), control (square) or Prrx1 (triangle) ASO; n=5 per group. (D) Dot plots with median showing the mRNA expression of Acta2, Col1a1 and Fn1 at day 7 in mice treated with either PBS (circle), Control (square) or Prrx1 (triangle) ASO; n=5 per group. (Scale bar in B: 80µm in low magnification images and 40µm in high magnification ones); (Abbreviations: Control (Cont), Antisense oligonucleotide (ASO)). Kruskal-Wallis test with Dunns post-test, *p≤0.05.

Would the decrease in Prxx1 expression lead to a change in the activity of the resident mesenchymal niche (Cd45-Cd31-EpCam-Sca1+ cells)? This can be easily tested using alveolospheres (co-culture of normal mature AT2s with either rMC coming from ASO control or ASO Prrx1 lungs). If the mesenchyme from ASO Prrx1 lung is more efficient in promoting alveolosphere formation and growth, this would argue that the cells are less prone to adopt a myofibroblast-like phenotype when Prrx1 is silenced.

We thank the reviewer for this valuable comment and suggestions. However, it is worth noting that our study used a pharmacological approach, which may have limited the duration and extent of *Prrx1* inhibition after isolation of the mesenchymal cells. We would like to highlight that this approach may not be suitable for long-term culture of lung spheroids, embedded in thick Matrigel. Matrigel could have also restricted the diffusion of additional ASO treatment in culture. Usually, mesenchymal cells with constitutive alterations, derived from transgenic models, would be more appropriate for this assay.

As a surrogate, we used instead wild-type or *Prrx1-/-* MEFs as stromal cells in a lung spheroid assay to investigate whether PRRX1 loss could perturb epithelial homeostasis and regeneration. The results, which we now mention in the revised manuscript (lines 371-375 and revised Figure 5—figure supplement 5), showed that *Prrx1* loss did not dampen but rather promoted epithelial spheroid formation. As suggested by the reviewer, these findings may suggest that *Prrx1-*deficient stromal cells, specifically MEFs in this case, may be less likely to adopt a myofibroblast-like phenotype when Prrx1 is silenced. The altered myofibroblastic differentiation of *Prrx1^-/-^* MEFs in response to TGF-β1 also supported this hypothesis (see Figure 5—figure supplement 5). While these findings are promising, it is important to note that further research using isolated lung mesenchymal cells after conditional loss of *Prrx1* in transgenic mice is necessary to confirm our results.

Supplemental Figure S2: PRRX1 expression profiles at single-cell resolution using the "IPF Cell Atlas" web database (http://ipfcellatlas.com/).Figure S2C: Increase in PRRX1 expression is observed in Myofibroblasts in IPF vs. Control as expected.However, there is also a significant increase in PRRX1 expression in the PLIN2+ fibroblasts in IPF vs. Control. What is the significance of this increase? Could it be that this leads to the loss of mesenchymal niche activity for AT2 cells (see comments for figure 7)?

The potential role of PRRX1 in regulating the phenotype of *PLIN2+* fibroblasts in lung fibrosis is a very interesting avenue as a “two-way conversion between lipogenic and myogenic fibroblastic phenotypes” was previously reported during lung fibrosis (Agha et al., Cell Stel Cell. 2017). In future studies, conditional loss using transgenic mouse models would be a more relevant approach to investigate the role of PRRX1 in lipofibroblasts and in the mesenchymal niche activity as suggested by the reviewer.

Hence, in the revised manuscript, we now acknowledge that: “previous studies have showed that PRRX1 TFs were involved in adipogenesis (Du et al., 2013) as well as potentially influencing the cell fate switch between alveolar myofibroblasts and lipofibroblasts during fetal lung mesodermal development (Li et al., 2016). Given the important role of lipofibroblasts in lung repair and epithelial regeneration (El Agha et al., 2017), further investigation is required to fully understand the significance of the observed increase in PRRX1 expression in this fibroblast subtype. Future studies could also explore the effects of targeting *Prrx1* loss specifically in these populations, including the fibrotic *Lrrc5^pos^* cluster, to provide a more comprehensive understanding of the role of PRRX1 in lung fibrosis” (see lines 517-526).

Supplemental Figure S10: PRRX1 is increased during the fibrotic phase in mice with bleomycin-induced fibrosis.What is happening to PRRX1 expression from d14 to d28 (during fibrosis resolution… Is PRRX1 expression decreasing)?

In response to the reviewer's comments, we extended the study of *Prrx1* expression levels after bleomycin challenge beyond day 14, namely to day 21 and day 28, as shown in Figure 6—figure supplement 1. As described in the revised manuscript (line 382-388), a slight decrease *in Prrx1b* mRNA expression and PRRX1 total protein levels was observed at day 21 compared to days 14 and 28 during the late fibrosis phase. However, PRRX1 transcription factors were still upregulated at the mRNA and protein levels on days 21 and 28 compared to baseline (see revised Figure 6-supplement figure 1).

Supplemental Figure S15: a summary sketch of PRRX1 regulation and functions in lung fibroblasts.The model is not clear as we would expect PRRX1 to first be upregulated in the context of TGFb1 signaling and then downregulated at a later stage due to the establishment of a negative feedback loop. In addition, PRRX1 should be located downstream of TGFBR2?

In response to the reviewer's suggestions, we have made modifications to the summary sketch in new figure 9. Specifically, we have made a clear separation between the fibroblast "phenotype" at baseline and after undergoing myofibroblastic differentiation in response to TGFb1. Additionally, we have taken into account the late inhibition of PRRX1 by TGFb1 at 48 hours as a potential negative feedback loop to modulate TGFb1 responsiveness. We hope that these changes improved the clarity and accuracy of our summary sketch and address the reviewer's concerns.